# Layered entrenchment maintains essentiality in the evolution of Form I Rubisco complexes

Luca Schulz [ID][1], Jan Zarzycki [ID][1], Wieland Steinchen[2,3], Georg K A Hochberg [ID][1,2,3 ✉] & Tobias J Erb [ID][1,2 ✉]

## Abstract

**Protein complexes composed of strictly essential subunits are abundant in nature and often arise through the gradual complex-ification of ancestral precursor proteins. Essentiality can arise through the accumulation of changes that are tolerated in the complex state but would be deleterious for the standalone complex components. While this theoretical framework to explain how essentiality arises has been proposed long ago, it is unclear which factors cause essentiality to persist over evolutionary timescales. In this work we show that the central enzyme of photosynthesis, ribulose-1,5-bisphosphate carboxylase/oxygenase (Rubisco), can easily start to depend on a newly recruited interaction partner through multiple, genetically distinct mechanisms that affect sta-bility, solubility, and catalysis. We demonstrate that layering multiple mechanisms of essentiality can lead to its persistence, even if any given mechanism reverts. More broadly, our work highlights that new interaction partners can drastically re-shape which substitutions are tolerated in the proteins they are recruited into. This can lead to the evolution of multilayered essentiality through the exploration of areas of sequence space that are only accessible in the complex state.**

**Keywords** Protein Complex Evolution; Evolutionary Biochemistry; Evolution of Essentiality; Layered Dependence; Rubisco
**Subject Categories** Evolution & Ecology; Plant Biology

## Introduction

Biological systems are replete with essential cellular components. A component is considered essential if some biological process or function cannot be carried out at all when the component in question is missing. How any component becomes essential for a living cell is an important question in evolutionary biology (Muller, 1918). Protein complexes whose functions strictly depend on the presence of several different subunits are a tractable example of this process (Ali and Imperiali, 2005). It seems unlikely that such complexes could develop in a single step, i.e., that all essential

components are recruited into a given complex at the same time. A simpler explanation is that this process occurs in two steps: First a new subunit is recruited into a pre-existing complex (or when a pre-existing subunit duplicates). Initially the pre-existing compo-nents retain their ability to function without the new subunit, but in a subsequent step they lose this ability (Schulz et al, 2022b; Force et al, 1999; Hochberg et al, 2020; Stoltzfus, 1999). The novel component is now 'entrenched', which means that the complex can no longer carry out its full function in the absence of the new subunit. In this framework, entrenchment is the evolutionary process that results in the biological phenomenon of essentiality.

A general and conceptually simple genetic mechanism for entrenchment has been known for over two decades. Entrenchment only requires the accumulation of conditionally tolerated substitu-tions in the interaction partner of the new component (Schulz et al, 2022b; Hochberg et al, 2020). That is, substitutions that are functionally deleterious in the absence of the novel component but tolerated when it is present (Fig. 1A) (Stoltzfus, 1999; Muñoz-Gómez et al, 2021). Once enough such substitutions have accumulated, the complex is no longer functional in the absence of the novel component. Another way to understand this mechanism is that novel subunits alter the functionally accessible sequence space of the components they interact with (Schulz et al, 2022b). Once a complex evolves into parts of sequence space that are only accessible in the presence of the new subunit, that subunit becomes essential (Fig. 1A). In the few model systems where the genetic basis of dependence has been worked out, empirical work has shown that just one conditionally tolerated substitution can be sufficient to render a protein completely dependent on a novel component (Finnigan et al, 2012; Schulz et al, 2022a). These studies have established that essentiality is not particularly difficult to evolve through the process of entrenchment.

What remains a problem is why essentiality is sometimes very long lasting (Emlaw et al, 2021; Hochberg et al, 2020). If a single substitution can entrench a novel subunit, a single reversion may also very well revert the entrenchment. There are in principle two mechanisms that can maintain essentiality over evolutionary timescales: the first is that purifying selection directly preserves substitutions that are responsible for entrenchment. This would be the case if these substitutions are functionally beneficial, for example because they enhance an enzyme's catalytic performance (Spreitzer, 2003; Joshi et al, 2015). However, no such functional advantage has been detected for examples in which single

[1]Department of Biochemistry & Synthetic Metabolism, Max Planck Institute for Terrestrial Microbiology, Karl-von-Frisch Straße 10, 35043 Marburg, Germany. [2]Center for Synthetic Microbiology (SYNMIKRO), Philipps University Marburg, Karl-von-Frisch Straße 14, 35043 Marburg, Germany. [3]Department of Chemistry, Philipps University Marburg, Karl-von-Frisch Straße 1, 35043 Marburg, Germany. ✉E-mail: georg.hochberg@mpi-marburg.mpg.de; toerb@mpi-marburg.mpg.de

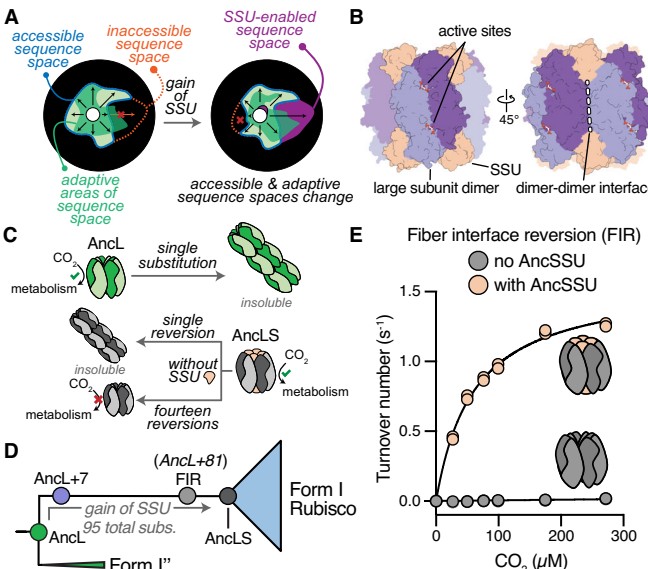

**Figure 1. Protein complex evolution during the emergence of Form I Rubiscos.**

(A) Schematic representation of a protein's (functional) sequence space and how the gain of complexity (here small subunit (SSU)) can modulate it. Accessible sequence space is shown in light green with blue outlines, inaccessible sequence space as an orange dotted line, functional/adaptive sequence space in green, and SSU-enabled sequence space in purple. (B) AncL+7 structure surface representation highlighting the approximate position of active sites, SSU-binding sites, the dimer-dimer interface, and catalytic large subunit dimers within the overall complex. (C) The ancestral AncL can become insoluble in a single historical substitution, whereas more than one reversion is required in the background of the derived AncLS to revert insolubility. Insolubility is circumvented by a combined fourteen substitution, which results in the FIR construct, which is catalytically inactive without AncSSU. (D) Schematic representation of the investigated evolutionary interval with AncL+7 and the fiber interface reversion (FIR) construct highlighted as hypothetical intermediates during the evolution from AncL to AncLS. (E) Activity of FIR without or with addition of a ten-fold AncSSU excess under varying dissolved $CO_2$ concentrations (see Appendix Table S1). Source data are available online for this figure.

substitutions entrenched interactions (Finnigan et al, 2012; Schulz et al, 2022a). In these cases, essentiality can persist if the likelihood of entrenching substitutions significantly exceeds the likelihood of entrenchment-reversing substitutions (Hochberg et al, 2020). In other words, the sequence space associated with entrenchment must be sufficiently large such that once entrenched, a complex is unlikely to drift back out of it and into parts of sequence space in which the novel subunit is no longer essential again.

Here we use the evolution of a new essential subunit in the protein complex of the $CO_2$-fixing enzyme Rubisco to understand the genetic and structural mechanisms that have maintained this interactions' essentiality for over 2 billion years of evolution. Modern-day Form I Rubiscos assemble into a complex of eight catalytic large subunits (LSUs) and eight non-catalytic, accessory small subunits (SSUs, L8S8 complex, Fig. 1B). Even though more ancestral forms of Rubisco exist and are functional without the SSU, all known Form I Rubiscos are strictly dependent on a SSU for solubility and activity. In prior work, we used ancestral sequence reconstruction to resurrect ancestral Rubiscos that existed before

and after the SSU first appeared (named AncL and AncLS, respectively, Fig. 1D) (Schulz et al, 2022a). AncLS was already totally dependent on the SSU for solubility and activity. We found that a complete dependence on AncSSU for solubility can be created by introducing a single historical substitution, which occurred along the interval that separates AncL from AncLS, into AncL. This substitution causes Rubisco to polymerize into insoluble fibers in the absence of AncSSU, but polymerization is prevented through steric hindrance when AncSSU is bound (Schulz et al, 2022a; Garcia-Seisdedos et al, 2017).

This system has two further qualities that make it uniquely attractive to study the persistence of essentiality. First, the substitutions that caused Rubisco to become dependent on the SSU for solubility do not appear to have improved Rubisco's performance when they first arose (Schulz et al, 2022a). It is therefore unlikely that the SSU's essentiality is maintained as a secondary effect of purifying selection acting on a functional benefit associated with the entrenching substitutions. And second, polymerization into fibrils is not the only mechanism of entrenchment: we succeeded in ridding AncLS of its dependence on the SSU for solubility by reverting amino acids at 14 sites back to their states in AncL (Fig. 1B–D). The resulting "revertant" enzyme was highly soluble, but not active in the absence of the SSU. This suggested the existence of a separate mechanism of essentiality that is genetically and biochemically distinct from fiber formation. Such layered entrenchment would preserve the essentiality of the SSU, even if the protein can drift in and out of any one mechanism of entrenchment.

Here we investigate the mechanism behind these deeper layers of Rubisco's dependence on AncSSU. We discover that at least two additional biochemically and genetically distinct mechanisms—stability and catalysis—can also create a dependence through a small number of historical substitutions. We find that catalytic entrenchment is easy to create, but very difficult to revert. Our results imply that the recruitment of the SSU changed Rubisco's accessible sequence space, which allowed Rubisco to accumulate substitutions that are tolerated in the presence of the SSU but would otherwise be deleterious. Over the course of evolution, this resulted in easily accessible, multilayered, and genetically complex entrenchment, which likely maintained the SSU's essentiality in all Form I Rubiscos, across more than 2 billion years of evolution in diverse lineages. More broadly, our study provides an example of how novel components can cause a drastic rewiring of a protein complex' accessible sequence space.

## Results

To investigate the evolution of Rubisco's layered dependence on AncSSU, we utilize two variants of our ancestral Rubiscos AncL and AncLS. The first is AncL+7. This is a version of the ancestor AncL into which we introduced 7 historical substitutions (i.e., amino acids present in AncLS at 7 sites) that allow it to interact with AncSSU without causing a dependence (Schulz et al, 2022a). We use this construct throughout to test which additional substitutions that separated AncL from AncLS cause a dependence on AncSSU for catalysis when introduced into AncL+7.

The second construct we use is what we call the fiber interface reversion (FIR) construct of AncLS. In this construct we have reverted amino acids at 14 sites back to their states in AncL. This abolished the

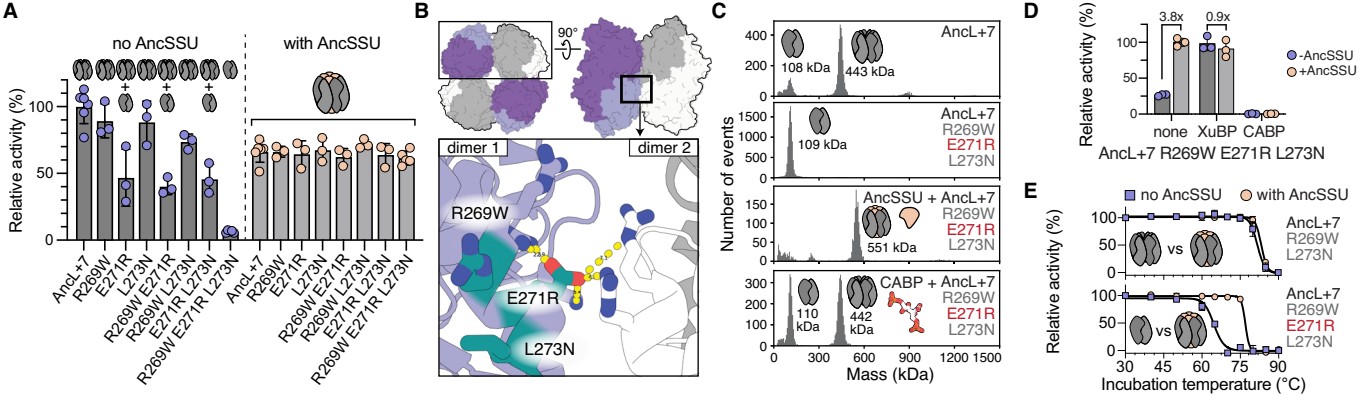

**Figure 2. Stability entrenchment of Rubisco's small subunit.**

(A) Relative activities of AncL+7 and single, double, and triple substitution constructs on the trajectory to AncL+7 R269W E271R L273N with and without the presence of a ten-fold AncSSU excess. Oligomerization state of the investigated variant at assay conditions is indicated schematically as cartoons above the bars. Activity is given relative to the activity of AncL+7 without AncSSU and measured at 25 °C by following the depletion of UV signal at 340 nm using a coupled enzyme assay described in the materials and methods section with slight adaptations (55 mM NaHCO₃ and 2.5 mM RuBP used). Mean of N = 6 (AncL+7), N = 5 (AncL+7 REL), or N = 3 (other variants) measurements depicted with error bars showing the standard deviation. (B) Localization of introduced substitutions in the structure of AncL+7 (PDB ID: 7QSX). Top shows a surface representation to localize the boxed cartoon representation. (C) Mass photometry (MP) measurements of AncL+7 and AncL+7 R269W E271R L273N (with or without AncSSU). Inferred oligomeric states are depicted as cartoons. Proteins were crash diluted from 20 μM stock concentration and measured at 50 nM final concentration, as well as MP measurement of AncL+7 R269W E271R L273N in presence of 0.2 mM CABP shows inhibitor-dependent oligomerization. (D) Relative activity of AncL+7 R269W E271R L273N activity assays under standard conditions ('none') or in the presence of 1 mM XuBP or 0.2 mM CABP. Activity is given relative to the activity of the AncSSU-bound triple substitution construct in absence of inhibitory sugars. Green datapoints/dark gray bars indicate measurements in absence of AncSSU, gray datapoints/light gray bars indicate measurements in presence of a five-fold AncSSU excess. Relative activities at 25 °C were determined by following the rate of NADH oxidation using a coupled enzyme assay described in detail in the materials and methods section. Mean of N = 3 measurements depicted with error bars showing the standard deviation. (E) Resistance to thermal denaturation of AncL+7 R269W L273N (remains octameric) and AncL+7 R269W E271R L273N (becomes dimeric without AncSSU), as assessed by incubating protein at varying temperatures for 1 h, prior to assessing remaining activities at 25 °C. Activity is given relative to the activity of the respective variant pre-incubated at 30 °C. Mean of N = 3 measurements depicted with error bars showing the standard deviation. Sigmoidal fit is depicted to guide the reader. Source data are available online for this figure.

dependence of AncL on AncSSU for solubility, but still left the resulting Rubisco dependent on AncSSU for catalytic activity (Schulz et al, 2022a). In the presence of AncSSU, this variant of AncLS is surprisingly active (Fig. 1E, 1.58 s⁻¹, compared to AncL+7's 0.63 s⁻¹, Appendix Table S1), which likely stems from a productive combination of historical substitutions that epistatically lead to high catalytic rates. We use this FIR construct to test if reverting entrenchment-causing substitutions from the first step is sufficient to make this construct fully independent of AncL. A comprehensive list and description of all variants is given in Appendix Table S2.

## Stability entrenchment

We first set out to find substitutions that could induce a dependence on AncSSU in AncL+7 without making the resulting Rubisco insoluble in the absence of AncSSU (Fig. 1C). We noticed that helix α4 (aa 261–274), which faces Rubisco's central pore, was heavily substituted from AncL to AncLS and that three sites within that region changed to amino acids that are well conserved in Form I Rubiscos (R269W, E271R, and L273N). Introducing these substitutions into AncL+7 yielded a highly soluble Rubisco variant (AncL+7 REL) that exhibited ~90% decreased activity without AncSSU in our assays (Fig. 2A; Appendix Fig. S1). Addition of AncSSU almost entirely recovered the activity of AncL+7 REL. This suggested that a catalytic dependence of Rubisco on AncSSU can be produced in few substitutions, even though the exact mechanism remained unclear at this stage.

We next sought to identify the biochemical basis of this dependence on AncSSU. The three substitutions (R269W, E271R, and L273N, abbreviated as REL) are localized in proximity to

Rubisco's dimer-dimer interface (Fig. 2B). We thus hypothesized that their introduction hinders octamer formation. Indeed, instead of forming an octamer, the AncL+7-based triple mutant AncL+7 REL was dimeric in the absence of AncSSU, as determined by mass photometry (MP, Fig. 2C). Even a single substitution (E271R) was sufficient to abolish octamer formation (Appendix Fig. S2) and led to ~60% reduced activity under assay conditions, whereas activity in presence of AncSSU was unchanged (Fig. 2A). The addition of AncSSU recovered octamerization and led to the formation of canonical L8S8 complexes (Fig. 2C). This indicates that binding of AncSSU can compensate for the detrimental effect of these substitutions by holding together these weaker octamers. Overall, this data suggested that AncSSU buffers substitution at the dimer-dimer interface, which destabilize oligomerization.

However, it is unclear if assembly into octamers is actually required for catalysis. Recent studies suggest that oligomerization is not crucial for activity in Form II Rubiscos (Liu et al, 2022), which acquired their higher-order assemblies (dimers, tetramers, and hexamers) independently from Form I Rubiscos. It is thus conceivable that the three REL substitutions had two separate effects on Rubisco: eroding the octamer interface and potentially rendering catalysis AncSSU-dependent independently of octamer formation. We thus tested if AncL+7 REL depends on AncSSU for activity because octamer formation is essential for function. To do this, we first measured how the addition of substrate and substrate analogs affected AncL+7 REL's assembly into oligomers. We could not observe substrate-dependent (RuBP) oligomerization, which has been reported for some Rubiscos (Gunn et al, 2017). We did, however, observe that the addition of known active site inhibitors

of Rubisco, namely xylulose-1,5-bisphosphate (XuBP) (Bracher et al, 2015) or 2-carboxyarabinitol-bisphosphate (CABP) (Parry et al, 2007), induced octamer formation in AncL+7 REL (Fig. 2C; Appendix Fig. S3). We additionally solved the crystal structures of AncL+7 REL in CABP-bound forms (with and without AncSSU), in which the protein was present exclusively as an octamer with or without AncSSU (PDB 8QMW and 8QMV, respectively, Appendix Table S3). A strong enough active site binder or very high concentrations of the protein can thus force AncL+7 REL into octamers, even though its native substrate RuBP cannot (Appendix Fig. S4). This implied that there is a direct link between Rubiscos active site and the dimer-dimer interface and suggested that Form I Rubiscos need to be octameric to be fully catalytically active.

We were able to test this theory directly because unlike with plant Rubiscos, XuBP does not inhibit catalysis in AncL+7-derived Rubiscos (Appendix Fig. S5). This enabled us to measure activity in the presence of XuBP while shifting Rubisco's oligomeric state toward the octamer. Adding 1 mM XuBP to activity assays of AncL+7 REL containing 3 mM RuBP indeed recovered activity in the absence of AncSSU (Fig. 2D). Notably, no decrease in catalytic rate was observed when 1 mM XuBP was added to both the activation and assay mixture (Appendix Fig. S5). This suggested to us that XuBP can still efficiently diffuse out of Rubisco's active site after inducing oligomerization, which would then allow for binding and turnover of RuBP and $CO_2$ before the octamer dissociates. Notably, the effect was XuBP-specific and could not be induced by non- or mono-phosphorylated xylulose or the tight-binding active site inhibitor CABP, which likely stays bound and inhibits catalysis (Appendix Figs. S5B and S6A). To independently test for octamer-dependence of catalysis, we crash diluted AncL+7 E271R from high protein concentrations (>100 μM), at which we hypothesized octamers to still be formed, to a normalized assay concentration. This resulted in recovered activity when initiating assays from high pre-dilution concentrations (Appendix Fig. S6B).

Together these results imply that Form I Rubiscos are only fully active as octamers, which facilitates the development of a dependence on AncSSU. Because AncSSU bridges adjacent Rubisco dimers, it can provide extra stability to the inter-dimer interface. This, in turn, allows the dimer-dimer interface between LSU dimers to degrade through the accumulation of destabilizing substitutions, which are compensated by the SSU at the cost of dependence.

In addition, we also noticed that AncL+7 REL (as well as all other constructs that form dimers instead of octamers without AncSSU) were drastically destabilized towards thermal denaturation (~30 °C), which was partially recovered by AncSSU addition (Fig. 2E; Appendix Fig. S7). This implies that LSUs in dimers are less stable than LSUs associated into octamers. In hot environments, this could potentially serve as separate mechanism (independent of octamer-dependent catalysis) through which a dependence on the SSU could have developed. Increased stability as an octamer could plausibly provide a direct selective advantage given the environmental context, or could offer evolution room for adaptive, yet destabilizing evolution (Taverna and Goldstein, 2002; Tokuriki and Tawfik, 2009).

## Catalytic entrenchment

Our observations on AncL+7 REL imply that AncSSU could become essential for catalysis because it provided stability to octamers. However, this cannot be the sole reason for AncLS' catalytic dependence on AncSSU: our FIR construct of AncLS (which led to

this investigation) was octameric, yet catalytically inactive without AncSSU (Fig. 1E). This suggested that there must be yet another mechanism for catalytic dependence on AncSSU, likely genetically and structurally independent from the ability to form octamers.

We could not rationally identify an obvious set of candidate substitutions for catalytic AncSSU-dependence from the sequence differences between AncL and AncLS. We therefore sought an alternative way to identify plausible sites. Based on our findings regarding higher oligomerization states in the presence of XuBP and CABP, we reasoned that sites distal from the active site that undergo changes upon substrate binding would be promising candidates in which substitutions could lead to AncSSU dependence (Bhattacharya et al, 2022).

To find such sites, we performed hydrogen/deuterium exchange mass spectrometry (HDX) on AncL+7 alone and AncL+7 bound to the transition state analog CABP (Appendix Fig. S8). HDX quantifies the time-dependent exchange of hydrogen to deuterium in proteins, which is dependent on solvent accessibility. When comparing conditions, differences in solvent accessibility can stem from structural re-arrangements or changes in protein motion. As expected, comparing HDX rates between the two conditions revealed (re-)ordering of the active site and regions known to be involved in catalysis, including loop 6 (Andersson and Backlund, 2008), regions in proximity to the central solvent channel (Spreitzer et al, 2005), and the 60 s loop (Taylor and Andersson, 1996), upon CABP inhibitor binding. Crucially, however, it also revealed regions up to ~15 Å from the active site that decreased their deuterium uptake upon inhibitor binding (Fig. 3A).

We chose to investigate the effect of ten substitutions in regions with the most drastic HDX response to inhibitor binding, many of which were outside of regions that were previously connected to catalytic activity in Rubisco (Fig. 3B). We first introduced these substitutions into AncL+7 as single substitutions. Three of the ten substitutions led to a 40-60% reduction in activity, relative to AncL+7 (Fig. 3C). One substitution of an internal leucine to isoleucine (L192I) reduced activity to ~40% of AncL+7. In the presence of AncSSU, activity of this construct only decreased by 13% ($P = 0.03$, two-tailed T-test), relative to the AncSSU-bound control of the parent variant AncL+7 (Fig. 3D). This indicates that the presence of AncSSU can buffer the detrimental effect of the L192I substitution on the catalysis of AncL+7. Another substitution, G158C, also impaired catalysis in the absence of AncSSU. We next combined L192I and G158C into a double mutant. This reduced the rate of catalysis by another ~60% relative to the L192I single mutant. The deleterious effect of this substitution was less detrimental in the presence of AncSSU, with only ~30% activity lost (Fig. 3D). Notably, both the AncL+7 L192I single and the AncL+7 G158C L192I double mutants retained their octameric assembly, indicating that a mechanism distinct to the loss of oligomerization was causing dependence on AncSSU for catalysis (Fig. 3E).

Next, we tested, whether L192I and G158C were the only two substitutions that contributed to the dependence of AncLS on AncSSU for catalysis. To do this, we reverted C158G and I192L in the soluble, yet catalytically inactive, FIR construct. This recovered ~25% of catalytic activity in the absence of AncSSU (Fig. 3F, $P = 0.00002$). This indicates that C158G and I192L contribute to the dependence of AncLS on AncSSU for catalysis, but equally that additional substitutions deepened its catalytic dependence on AncSSU further. Like other dependencies, the need to bind an interaction partner for catalytic activity can thus be created in a

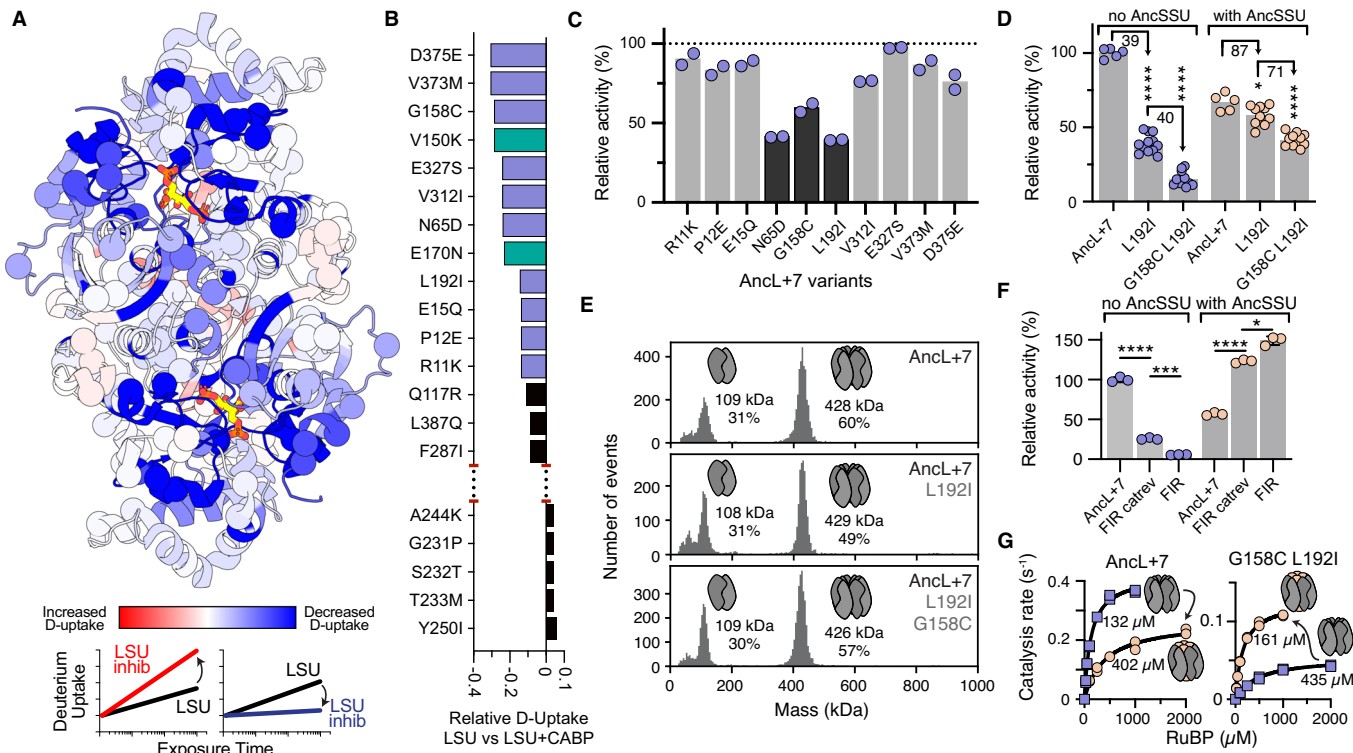

**Figure 3. Catalytic entrenchment of Rubisco's small subunit.**

(A) Relative hydrogen-deuterium exchange mass spectrometry (HDX) deuterium uptake rates of AncL+7 with versus without inhibitory sugar (CABP) measured 95 s after protein dilution into deuterated buffer and projected onto a catalytic dimer of AncL+7 (PDB ID: 7QSX). Scheme at bottom indicates the expected deuterium uptake behavior of regions with increased and decreased uptake rates. (B) Substituted sites with the highest and lowest relative D-uptake rate differences when comparing AncL+7 versus AncL+7 with CABP in ranked form. Investigated substitutions from AncL to AncLS are indicated in purple (this study) or teal (previous work (Schulz et al, 2022a)). (C) Relative activity of AncL+7-based single substitution constructs, relative to the activity of AncL+7 without AncSSU. Relative activities at 25 °C were determined by following the rate of NADH oxidation using a coupled enzyme assay described in detail in the material and methods section. Mean of $N = 2$ measurements depicted with individual datapoints shown. (D) Relative activity of AncL+7 single and double substitution constructs with and without presence of a five-fold AncSSU excess. Activity is relative to the activity of AncL+7 without AncSSU. Significance was tested by two-tailed t test between AncL+7 and the variant for which the significance is reported (P values: (1) = < 0.000001, (2) = < 0.000001, (3) = 0.029437, (4) = 0.000001). *$P < 0.1$; ****$P < 0.00001$. Percentage residual activity between relevant variants is indicated above the respective bars. Mean of $N = 5$ (AncL+7), $N = 12$ or 10 (AncL+7 L192I with and without AncSSU, respectively), and $N = 11$ (AncL+7 G158C L192I, with and without AncSSU, respectively) measurements depicted with individual datapoints shown. Relative activities were determined as described for Fig. 3C. (E) Mass photometry measurements of AncL+7 and relevant single or double substitution constructs. Inferred oligomeric state is shown schematically, measured mass is listed, and the total percentage of binned events relative to all events of the measurement are listed. (F) Relative activity of AncL+7 and the fiber interface reversion (FIR) construct, as well as the FIR construct with "G158C" and "L192I" reverted to their state in AncL+7 (catrev) measured in absence (blue) and presence (teal) of a ten-fold AncSSU excess. Relative activities were determined as described for Fig. 3C and activity is given relative to the activity of AncL+7 without AncSSU. Significance was tested by two-tailed t test between the indicated variants (P values: (1) = 0.000003, (2) = 0.000018, (3) = 0.000002, (4) = 0.001576). *$P < 0.01$; ***$P < 0.0001$; ****$P < 0.00001$. Mean of $N = 3$ measurements depicted with error bars showing the standard deviation. (G) RuBP Michaelis Menten kinetics for AncL+7 and AncL+7 G158C L192I without (purple) and with (beige) a five-fold AncSSU excess. Inset values list the measured $K_m$(RuBP) in presence of AncSSU. Catalysis rate at 25 °C was determined by following the rate of NADH oxidation using a coupled enzyme assay described in detail in the methods section. Individual datapoints ($N = 2$) shown. Source data are available online for this figure.

small number of substitutions but requires a larger set of reversions to abolish. Interestingly, reverting C158G and I192L to their non-entrenching states slightly, yet significantly ($P = 0.0016$), decreased the activity of the FIR construct in the presence of AncSSU (Fig. 3F). This suggests that substitutions which deepen the dependence on AncSSU can simultaneously be functionally beneficial for the AncSSU-bound state.

To follow up on this observation, we investigated whether the G158C and L192I substitutions were beneficial to catalysis, as suggested above, or functionally neutral in the same way that the fiber entrenching substitution was. In AncL+7 the $K_m$(RuBP) increased roughly 3-fold upon AncSSU addition ($132 \pm 14\,\mu M$ to $402 \pm 90\,\mu M$), whereas this effect was inverted in AncL+7 G158C L192I and $K_m$(RuBP) decreased from $435 \pm 102\,\mu M$ to $161 \pm 16\,\mu M$

(Fig. 3G; Appendix Table S1). While this positive effect on $K_m$ still coincided with an overall decreased catalytic rate ($0.13 \pm 0.01\,s^{-1}$ (AncL+7 G158C L192I with AncSSU) versus $0.26 \pm 0.02\,s^{-1}$ (AncL+7 with AncSSU)), this change in kinetic parameters might be/have been beneficial, given the right intracellular conditions. This is circumstantial evidence that conditionally tolerated substitutions can be advantageous. These results imply that novel subunits might open adaptive evolutionary paths that would otherwise not be accessible.

## Discussion

Using ancestral Rubisco's dependence on AncSSU as an example, we demonstrate a potentially more general principle that explains

the persistence of essentiality beyond previously identified mechanisms, such as a strong mutational propensity towards entrenching states (e.g., amino acid states that favor hydrophobic entrenchment) (Hochberg et al, 2020): Novel interactions can change the accessible sequence space of proteins, thereby opening up many new genotypes that are only functional in the presence of the new component, and that are rooted in different effects (stability, as well as catalytic activity). In the case of Rubisco, these effects also encompassed positions that were far from the AncSSU binding site, which then enabled the development of a multilayered dependence on the new interaction. We can currently only speculate about the exact biochemical mechanisms behind these effects. Crystal structures of AncL+7 with and without AncSSU show only minor differences in Rubisco's ground state structure (Schulz et al, 2022a; Andersson and Backlund, 2008). We therefore believe that AncSSU binding alters Rubisco's dynamic motion on functionally relevant timescales across the protein, which in turn changes the effect of substitutions at sites with modified dynamics. Future work will have to test this prediction empirically.

Our work shows that each mechanism of entrenchment we discovered can be established through a small number of substitutions in AncL, but reverting the corresponding substitutions in AncLS is not enough to yield an AncSSU-independent, fully active Rubisco. This implies that there must be more entrenching states and likely additional mechanisms that remain to be discovered. One problem is that we do not know the temporal order of the substitutions that separate AncL from AncLS. Whether or not certain substitutions cause dependence on AncSSU is dictated by their exact sequence context. For example, structural entrenchment through substitutions that abolish oligomerization seems context-dependent in this way: the FIR construct contains states that when introduced into AncL+7 make it fall apart into a dimer, however, it nonetheless forms octamers in the absence of AncSSU. Additionally, the FIR construct is more active than ancestral variants with more historical substitutions introduced (e.g., $1.58 \, \text{s}^{-1}$ for the FIR construct (see Appendix Table S1), versus $0.29 \, \text{s}^{-1}$ for AncLS), which highlights the context dependence and epistatic interplay of historical substitutions. Still, we have been able to alleviate AncSSU dependencies of solubility and catalysis either fully or partially, in AncLS by reverting substitutions that were causal when introduced into AncL. This gives us confidence that the genetic and structural mechanisms we have discovered were relevant for Rubisco's historical evolution.

Form I Rubiscos have remained dependent on the SSU over more than 2 billion years of evolution, even though selection presumably does not act directly to maintain the SSU's essentiality. The multiplicity and depth of entrenchment mechanisms we discovered in our ancestors provides an explanation as to why: together these different mechanisms ensures that the SSU stays essential even if Rubisco drifts out of any one entrenching mechanism. This has happened at least once, in a cyanobacterial Rubisco that is partially soluble without the SSU (and hence has reduced its dependence on the SSU for solubility) but is barely active in the absence of the SSU (Andrews, 1988).

The SSU is not the only novel component Form I Rubiscos recruited in their history. Several dedicated assembly chaperones became completely essential for Rubiscos within the plant lineage (Aigner et al, 2017). Along the same lineage, Rubiscos additionally started to become dependent on Rubisco activases (Portis, 2003, 1995)—dedicated ATPases that remove inhibitory sugar phosphates from its active site. It is tempting to speculate that each addition enabled Rubisco to access new catalytic optima that reside in previously inaccessible parts of sequence space, which were opened by the addition of a novel component (Tokuriki and Tawfik, 2009). Alternatively, it is possible that Rubisco became addicted to such components without any functional benefit, solely because they unlocked a vast enough area of sequence space from which Rubisco has since simply failed to escape.

## Methods

### Reagents and tools table

| Reagent/Resource | Reference or Source | Identifier or Catalog Number |
|---|---|---|
| **Experimental Models** | | |
| *E. coli* NEB Turbo | New England Biolabs | C2984H |
| *E. coli* BL21 (DE3) | Thermo Fisher Scientific | EC0114 |
| *E. coli* ArcticExpress (DE3) | Agilent Technologies | 230192 |
| Additional Information | This Study | Appendix Table S6 |
| **Recombinant DNA** | | |
| pET16b | Merck Chemicals/ Novagen | 69662 |
| pET28b | Merck Chemicals/ Novagen | 69865 |
| pET16b-AncL+7 | (Schulz et al, 2022a) | N/A |
| pET16b-fiber interface reversion | (Schulz et al, 2022a) | N/A |
| pET28b-AncSSU | (Schulz et al, 2022a) | N/A |
| Additional Information | This Study | Appendix Table S5 |
| **Oligonucleotides and other sequence-based reagents** | | |
| PCR primers | This Study | Appendix Table S4 |
| **Chemicals, Enzymes and other reagents** | | |
| NaH$^{14}$CO$_3$ | Hartmann Analytics | ARC0138A |
| K$^{14}$CN | Hartmann Analytics | ARC0136A |
| D-[2-$^3$H] glucose | Hartmann Analytics | ART0112 |
| D-Ribose-5-phosphate disodium salt | Sigma-Aldrich | R7750 |
| Creatine phosphokinase | Sigma-Aldrich | CK-RO |
| Rabbit muscle aldolase | Sigma-Aldrich | A2714 |
| DL-α-Glycerol phosphate magnesium salt hydrate | Sigma-Aldrich | 17766 |
| Dihydroxyacetone phosphate lithium salt | Sigma-Aldrich | 37442 |
| Hexokinase | Sigma-Aldrich | 1142636001 |
| Glucose-6-phosphate dehydrogenase | Sigma-Aldrich | G5760 |
| 6-phosphate gluconate dehydrogenase | Sigma-Aldrich | P4553 |

| Reagent/Resource | Reference or Source | Identifier or Catalog Number |
|---|---|---|
| 2x Phusion High-Fidelity PCR Master Mix | New England Biolabs | M0531L |
| T4 DNA Ligase | New England Biolabs | M0202L |
| T4 Polynucleotide Kinase | New England Biolabs | M0201L |
| DpnI | New England Biolabs | R0176L |
| Protino Ni-NTA Agarose | Macherey-Nagel | 745400 |
| 4–20% Mini-PROTEAN® TGX™ Precast Protein Gels | Bio-Rad | 4561096 |
| 3-phosphoglyceric phosphokinase | Sigma-Aldrich | P7634 |
| Glyceraldehyde 3-phosphate dehydrogenase | Sigma-Aldrich | G2267 |
| Carbonic anhydrase | Sigma-Aldrich | C2624 |
| CO2/O2 gas mixtures | Air Liquide | Custom mixtures |
| Microscope coverslips | Carl Roth | LH25.2 |
| CultureWell™ Reusable Gaskets | Grace Bio-Labs | CW-50R-1.0 |
| Ribulose-1,5-bisphosphate (RuBP) | This Study | See Methods |
| Xylulose-1,5-bisphosphate (XuBP) | This Study | See Methods |
| 2-carboxyarabinitol-1,5-bisphosphate (CABP) | This Study | See Methods |
| [1-$^3$H]-RuBP | This Study | See Methods |
| Glycoaldehyde phosphate barium salt | This Study | See Methods |
| Sodium meta-periodate | Sigma-Aldrich | S1878 |
| AG 50W-X8 resin | Bio-Rad | 1421441 |
| PD-10 desalting columns | Cytiva | 17085101 |
| PRK from *S. elongatus* PCC 6803 | Oliver Mueller-Cajar, NTU Singapore | N/A |
| Phosphoriboisomerase from *Arabidopsis thaliana* | Oliver Mueller-Cajar, NTU Singapore | N/A |
| Additional Information | See Methods of this study | |
| **Software** | | |
| AcquireMP | Refeyn | N/A |
| DiscoverMP | Refeyn | N/A |
| ProteinLynx Global SERVER | Waters | N/A |
| DynamX | Waters | N/A |
| Phenix | Adams et al, 2010 | N/A |
| Coot | Emsley and Cowtan, 2004 | N/A |
| PyMOL 2.4.1 | pymol.org | |
| Prism 8 | graphpad.com | |
| Additional Information | See Methods of this study | |
| **Other** | | |
| ISERA SERAgel Column | Isera.de | Custom column |
| HPX-87H column | Bio-Rad | 1250140 |
| TwoMP mass photometer | Refeyn Ltd | N/A |
| ACQUITY UPLC M-Class System with HDX Technology | Waters | N/A |

| Reagent/Resource | Reference or Source | Identifier or Catalog Number |
|---|---|---|
| Enzymate BEH Pepsin Column (5 µm, 2.1 × 30 mm) | Waters | N/A |
| G2-Si HDMS mass spectrometer with ion mobility separation | Waters | N/A |
| Agilent 1260 Infinity Series HPLC | Agilent | N/A |
| Additional Information | See Methods of this study | |

## Chemicals and reagents

Chemicals were of the highest commercially available purity from Sigma-Aldrich and Carl Roth. NaH$^{14}$CO$_3$, K$^{14}$CN, and D-[2-$^3$H] glucose was obtained from Hartmann Analytics (Germany). Biochemicals (for cloning and protein production/purification) were obtained from Thermo Fisher Scientific, Macherey-Nagel, and New England Biolabs. 3-phosphoglyceric phosphokinase and glyceraldehyde 3-phosphate dehydrogenase were obtained from Sigma-Aldrich.

Rubisco-specific chemicals (D-Ribulose-1,5-bisphosphate (RuBP), Xylulose-1,5-bisphosphate (XuBP), 2-carboxyarabinitol-1,5-bisphosphate (CABP), and [1-$^3$H] RuBP) were synthesized following published protocols (Pierce et al, 1980; Kane et al, 1994; Bracher et al, 2015) as described below.

Non-radioactive RuBP was synthesized from D-Ribose-5-phosphate disodium salt (Sigma #R7750, 1.28 g) in a 400 mL reaction containing 40 mM MgCl2, 0.25 mM ATP, 15.2 mM creatine phosphate, 5 mM dithiothreitol, 6.5 mg creatine phosphokinase (Sigma #CK-RO), 6 mg *Arabidopsis thaliana* phosphoriboisomerase, and 4.6 mg *Synechocystis* sp. PCC 6803 phosphoribokinase. RuBP was purified using an XK 50/70 AG1-X8(Cl) column equilibrated to 3 mM HCl, 100 mM NaCl and eluted with a 6.25 column volume gradient increasing in salt concentration to 3 mM HCl, 250 mM NaCl. Collected fractions were lyophilized until dry, stored at −80 °C until further use, and subsequently dissolved in 50 mL of 3 mM HCl. The dissolved sample was desalted using a Sephadex G-10 XK26/100 column equilibrated and operated with 3 mM HCl. Collected fractions were pooled, aliquoted, and used directly after determining the RuBP concentration using a photospectrometric coupled assays described below.

XuBP was synthesized by aldol condensation of dihydroxyacetone phosphate with glycoaldehyde phosphate catalyzed by rabbit muscle aldolase (Sigma #A2714). Glycoaldehyde phosphate barium salt was synthesized in an aqueous 150 mL reaction by partially dissolving 12 mmol (2.33 g) glycerol phosphate magnesium salt hydrate (Sigma #17766) and 11 mmol sodium meta-periodate (2.35 g). The pH of the solution was adjusted to 6.0 with 1 M HCl followed by incubation at 37 °C for 1 h. Afterwards, 2 mmol glycerol was added to quench the reaction and the pH was adjusted to 7 with 1 M NaOH. 22 mmol (4.58 g) barium chloride was added and the reaction was incubated on ice for 1.5 h. Precipitated BaIO$_3$ was removed by filtration through a 0.22 µm sterile filter, the filtrate was mixed with four volumes (600 mL) absolute ethanol and stirred for 15 min to precipitate the glycoaldehyde phosphate barium salt. The precipitate was collected by centrifugation at 5000 × *g* for 20 min at 25 °C and subsequently dried by lyophilization.

For the aldolase-catalyzed synthesis of XuBP, 52.3 mg glycoaldehyde phosphate barium salt and 24 mg dihydroxyacetone phosphate (Sigma #37442) were partially dissolved in 2.5 mL 10 mM MES-NaOH pH 6.5 and 30 U rabbit muscle aldolase was added to initiate the reaction. The reaction was incubated at 37 °C for 3 h. Over the course of the reaction a white precipitate of XuBP-$Ba_2$ formed. After reaction completion, the XuBP-$Ba_2$ precipitate was collected by centrifugation at $3200 \times g$ for 15 min. The visibly distinct white XuBP-$Ba_2$ precipitate was separated from precipitate that was present from the onset of the reaction.

The white XuBP-$Ba_2$ precipitate, as well as the soluble reaction supernatant, were acidified to pH 3.0 via the slow addition of activated AG 50WX8-HCl-resin (~120 μL), which dissolved the precipitate. The resin was removed by filtration through a 0.45 μm syringe filter and the syringe was washed with 1 mL 3 mM HCl. The soluble XuBP filtrate was further purified by separation on a HiPrep Q HP 16/10 column equilibrated to 3 mM HCl with a gradient to 3 mM HCl, 350 mM NaCl over 10 column volumes (elution at ~125 mM NaCl). Fractions were assayed for the presence of XuBP using a photospectrometric coupled assay based on rabbit muscle aldolase and NADH oxidation via the activity of glycerol phosphate dehydrogenase. Fractions were pooled, the XuBP concentration was determined using a photospectrometric depletion assay, and fractions were stored at −80 °C until further use.

[1-³H] RuBP was synthesized enzymatically from D-[2-³H] glucose. Reactions were assembled in 1.7 mL containing 33 mM bis-tris-propane-HCl pH 7.5, 10 mM $MgCl_2$, 0.5 mM ATP, 2 mM creatine phosphate, 1.5 mM NADP, 10 mM dithiothreitol, 34 μg creatine kinase (Sigma #10127566001), 6.8 U Hexokinase (Sigma #1142636001), 6.8 U glucose-6-phosphate dehydrogenase (Sigma #G5760), 6.8 U 6-phosphate gluconate dehydrogenase (Sigma #P4553), 0.85 mg phosphoribulokinase from *Synechocystis* sp. PCC6803, 250 μL ³H-glucose (evaporated overnight at room temperature to a volume of ~25 μL), and 0.6 mM glucose (non-radioactive). Reactions were incubated for 4 h at 25 °C, 300 rpm shaking in a tabletop shaker. Subsequently, 250 μL crude reaction aliquots were stored at −80 °C and used as such for specificity assays.

¹⁴C-containing CABP was synthesized by using 1 mCi $K^{14}CN$ per synthesis reaction. At 55.5 mCi/mmol, 18.2 μmol radioactive $K^{14}CN$ was combined with 14.6 μmol RuBP. To this end, 431 μL of a 36 mM RuBP stock dissolved in 3 mM HCl was diluted with 200 μL 1 M tris-acetate pH 8.5 and the pH was adjusted to 8.3 using 1 M tris (free base). The buffered RuBP was added to the $K^{14}CN$ vial and left to react at room temperature for ~48 h. Crude reactions were purified via gravity Dowex 50W-X8(H⁺) columns with 1.8 mL resin as the bed volume. Resins were washed and equilibrated with 10 mL distilled $H_2O$ prior to application of the reaction to the resin. The flow-through, as well as $2 \times 1$ mL ddH₂O elution fractions, were collected, pooled, and subsequently dried under a gentle stream of $N_2$. Dried reaction product was dissolved in 8 mL of 50 mM Bicine-NaOH pH 9.3, aliquoted, and the radioactivity/specific activity was determined by counting aliquots/dilutions of the synthesized carboxypentinol-1,5-bisphosphate mixture. Fresh aliquots of CPBP were incubated overnight at 4 °C prior to first use to allow for de-lactonization.

## Molecular cloning and vector construction

All primers were obtained from Eurofins genomics. A list of all primers is provided in Appendix Table S4. Constructs destined to carry an N-terminal His-tag were cloned into the standard *E. coli* expression vector pET16b (Merck Chemicals) and those destined to carry a C-terminal His-tag were cloned into the standard *E. coli* expression vector pET28b (Merck Chemicals). Genes were amplified using 2x Phusion High-Fidelity PCR Master Mix (Thermo Fisher) with primers that introduce complementary overhangs to the respective PCR-linearized vectors. PCR products were used to construct the desired vectors using home-made 1.33x Gibson assembly master mix (Gibson et al, 2009). Assembly was verified by DNA sequencing (MicroSynth). All plasmids in this study are listed in Appendix Table S5.

Single-site mutants, insertions, truncations, and deletions were created using an adapted Q5 Site-Directed Mutagenesis procedure. Primers were designed using NEBasechanger (nebasechanger.neb.com) and used to amplify the entire desired vector with 2x Phusion High-Fidelity PCR Master Mix. PCR products (0.5 μL straight from the PCR) were used in reactions containing 2.5 μL ddH₂O, 0.5 μL T4 DNA Ligase (NEB #M0202L), 0.5 μL T4 DNA Ligase (NEB #M0202L), 0.5 μL T4 Polynucleotide Kinase (NEB #M0201L), and 0.5 μL DpnI (NEB #R0176L) for 2 h at 25 °C. Subsequently, the entire reaction was transformed into NEB Turbo Competent *E. coli* (NEB #C2984H, list of strains in Appendix Table S6) before the resulting vector was purified and mutagenesis success was verified by sequencing (MicroSynth).

## Protein production and purification

To produce proteins, plasmids encoding the respective genes were transformed into chemically competent *E. coli* BL21(DE3) cells and grown overnight on selective LB agar plates containing either 100 μg mL⁻¹ ampicillin or 50 μg mL⁻¹ kanamycin at 37 °C. Grown colonies were used to inoculate expression cultures in terrific broth (TB) medium containing the respective antibiotics. Expression cultures were grown in a shaking incubator at 37 °C to an $OD_{600}$ of 0.5–1.0, cooled down to 25 °C, induced with 0.5 mM isopropyl-β-D-thiogalactoside (IPTG), and subsequently left to produce protein overnight.

Cells were harvested by centrifugation at $8000 \times g$ for 10 min at 10 °C and cell pellets that were not used immediately were stored at −20 °C until further use. For purification, cell pellets were resuspended in buffer A (50 mM HEPES-NaOH, 500 mM NaCl, pH 7.6). Resulting suspensions were lysed using a Sonopuls GM200 sonicator (BANDELIN Electronic) at an amplitude of 55% with three consecutive cycles of 30 pulses over 60 s and the crude lysates were clarified by centrifugation at $13,300 \times g$ and 4 °C for 1 h. Clarified lysates were filtered (0.45 μm syringe tip filters) and applied to pre-equilibrated Protino Ni-NTA Agarose (Macherey-Nagel) beads in a gravity column. After loading the resin was washed with 45 column volumes of 15% (v/v) buffer B (50 mM HEPES-NaOH, 500 mM NaCl, 500 mM imidazole, pH 7.6) in buffer A before protein was eluted with 8 column volumes of 100% buffer B. The eluate was concentrated to a total volume of 2.5 mL and desalted using PD-10 desalting columns (GE Healthcare) and desalting buffer (25 mM Tricine-NaOH, 75 mM NaCl, pH 8.0). For crystallization and large-scale purifications, protein was further purified via size exclusion chromatography (SEC) on a Superdex 200 pg, HiLoad16/600 column (GE Healthcare). Elution fractions containing pure protein were determined via SDS-PAGE analysis on a 4–20% gradient gel (Bio-Rad), pooled, and

concentrated using 50 kDa MWCO centrifugal filters (Amicon). Purified Rubiscos in desalting buffer were used for crystallization immediately or stored at −20 °C in desalting buffer until further use. For crystallization, a ~2-fold molar excess of CABP (dissolved in 100 mM Bicine, 17.6 mM MgCl$_2$, pH 8.0) relative to the Rubisco concentration (determined by absorption at 280 nm) was added. Rubisco was carbamylated in a 3% (v/v) CO$_2$ atmosphere at 30 °C for 1 h and used for setting crystal plates immediately (see "Crystallization and structure determination").

## Thermal stability assay

Apparent melting temperatures of Rubisco were determined as published previously (Karkehabadi et al, 2005; Joshi et al, 2015). In short, 20 µM purified Rubisco was activated in 50 mM HEPES-NaOH (pH 8.0), 20 mM MgCl$_2$, 40 mM NaHCO$_3$, and 0.02 mg/mL carbonic anhydrase from bovine erythrocytes (Sigma-Aldrich) for 15 min at 25 °C. Activated Rubisco was incubated for 60 min at varying temperatures from 30 to 90 °C prior to a 5 min incubation on ice and a 1:10 dilution into assay mixtures. Activity assays contained a final concentration of 100 mM HEPES-NaOH (pH 8.0), 0.8 mM NADH, 10 mM MgCl$_2$, 0.5 µM 3-phosphoglyceric phosphokinase (pgk), 0.5 µM glyceraldehyde 3-phosphate dehydrogenase (gapdh), 5 mM ATP, 45 mM NaHCO$_3$, 0.02 mg/mL carbonic anhydrase and 10x diluted activation mixture (~2 µM activated Rubisco). Assays were initiated by the addition of 2 mM RuBP, and reaction progress was followed in a microplate reader (Tecan) at 25 °C by measuring the consumption of NADH, as determined by decreasing absorbance at 340 nm. Activities were normalized to the activity of Rubisco incubated at 30 °C.

## Radiometric kinetic analysis and CO$_2$/O$_2$ specificity assays

$^{14}$CO$_2$ fixation assays were carried out at 25 °C in 0.5 mL filled into 7.7 mL septum-capped glass scintillation vials (Wilson et al, 2018). Assay buffer (100 mM EPPS-NaOH (pH 8.00), 20 mM MgCl$_2$, 1 mM EDTA) and other required components were pre-equilibrated with CO$_2$-free N$_2$ gas. Beyond assay buffer, reactions contained an additional 0.01 mg/mL carbonic anhydrase, 2.2 mM self-synthesized RuBP and 5–70 mM NaH$^{14}$CO$_3$ (corresponding to ~100 to 800 µM $^{14}$CO$_2$). Dissolved CO$_2$ concentrations were calculated using the Henderson-Hasselbalch equation with pK values for carbonic acid of pKa1 = 6.25 and pKa2 = 10.33, while accounting for assay volume and headspace volume. Active site contents of purified Rubiscos were quantified for each data set by performing [$^{14}$C]-2-CABP binding assays on 0.2 nmol purified Rubisco, followed by separation from free ligand by size exclusion chromatography (Kubien et al, 2011) (Isera SEAgel-Column, custom made for HPLC) and active site quantification by scintillation counting. Purified Rubisco (~5–30 µM active sites) was activated in assay buffer supplemented with 50 mM NaHCO$_3$ and 20 µL of the activation mixture was used to initiate the assay. Reactions were carried out at 25 °C for 2 min before quenching with 200 µL 50% (v/v) formic acid. The specific activity of NaH$^{14}$CO$_3$ was determined by completely turning over 22.6 nmol RuBP using the highest employed NaH$^{14}$CO$_3$ concentration in a 60 min reaction.

Specificity assays were performed as published previously with slight adaptations (Kane et al, 1994; Wilson et al, 2018). In short, purified Rubisco was incubated in 20 mL septum capped glass

scintillation vials containing 1 mL 30 mM triethanolamine (pH 8.30), 15 mM Mg-acetate, and 0.01 mg/mL carbonic anhydrase. Assays were equilibrated in defined gas mixtures containing either 995,000 ppm O$_2$ and 5000 ppm CO$_2$ (for low specificity variants) or 999,291 ppm O$_2$ and 709 ppm CO$_2$ (for high specificity variants, Air Liquide, Germany) before starting reactions by addition of [1-$^3$H]-RuBP. Reactions were incubated for 60 min under constant gassing and subsequently dephosphorylated using alkaline phosphatase (10 U/reaction), separated on an Agilent 1260 Infinity series HPLC using a HPX-87H column (Bio-Rad), and relative amounts of glycerate and glycolate quantified by flow-through scintillation counting. Specificity was calculated as described previously (Kane et al, 1994).

## Ribulose-1,5-bisphosphate Michaelis-Menten kinetics

To determine the $K_M$(RuBP), photospectrometric reactions were set up with slight adaptations to previously published protocols (Kubien et al, 2011). Reactions were carried out at 25 °C in 100 mM HEPES-KOH (pH 8.00) and contained 10 mM MgCl$_2$, 2.5 mM ATP, 0.3 mM NADH, 2.5 U/mL pgk, 5 U/mL gapdh, 0.02 mg/mL carbonic anhydrase, 55 mM NaHCO$_3$, varying amounts of RuBP (0–5 mM), 0.2–2 µM activated Rubisco, and a 5-fold SSU excess, if applicable. Rubisco was activated by a 30-min pre-incubation in 50 mM HEPES-KOH (pH 8.00), 10 mM MgCl$_2$, 20 mM NaHCO$_3$, and 0.02 mg/mL carbonic anhydrase. Reaction progress was followed by measuring the depletion of NADH at 340 nm (Abs$_{340}$).

## Measurement of relative Rubisco activities

For head-to-head comparisons of Rubisco activities, photospectrometric reactions were set up with slight adaptations to the assays used to determine Ribulose-1,5-bisphosphate Michaelis-Menten kinetics and Rubisco inhibition assays. Reactions were carried out at 25 °C in 100 mM HEPES-KOH (pH 8.00) and contained 10 mM MgCl$_2$, 2.5 mM ATP, 0.3 mM NADH, 2.5 U/mL pgk, 5 U/mL gapdh, 0.02 mg/mL carbonic anhydrase, 70 mM NaHCO$_3$, 3 mM RuBP, 2 µM activated Rubisco, and a 5-fold SSU excess, if applicable. Rubisco was activated by a 30-min pre-incubation in 50 mM HEPES-KOH (pH 8.00), 10 mM MgCl$_2$, 20 mM NaHCO$_3$, and 0.02 mg/mL carbonic anhydrase. Reaction progress was followed by measuring the depletion of NADH at 340 nm (Abs$_{340}$).

## Mass photometry

Mass photometry measurements were carried out on microscope coverslips (1.5 H, 24 × 50 mm, Carl Roth) with CultureWell™ Reusable Gaskets (CW-50R-1.0, 50-3 mm diameter × 1 mm depth) that had been washed three times with distilled H$_2$O and 100% isopropanol and dried under a stream of pressurized air. Gaskets were assembled on microscope coverslips and placed on the stage of a TwoMP mass photometer (MP, Refeyn Ltd, Oxford, UK) with immersion oil. Samples were measured in 1x phosphate-buffered saline (PBS, 10 mM Na$_2$HPO$_4$, 1.8 mM KH$_2$PO$_4$, 137 mM NaCl, 2.7 mM KCl (pH 7.4)). To this end, 18 µL 1x PBS was used to focus the MP before 2 µL sample (0.5 µM protein) was added, rapidly mixed, and measured.

Samples were prepared by diluting purified protein to 20 µM monomer concentration in desalting buffer (25 mM Tricine (pH 8.00), 75 mM NaCl), as determined by absorption at 280 nm.

Immediately prior to measuring, samples were further diluted to a final concentration of 0.5 μM. For samples containing purified SSU, a 5-fold SSU excess was added to the undiluted sample shortly before dilution to 0.5 μM Rubisco monomer concentration.

Data was acquired for 60 s at 100 frames per second using AcquireMP (Refeyn Ltd, Oxford, UK). MP contrast was calibrated to molecular masses using 50 nM of an in-house purified protein mixture containing complexes of known molecular mass. MP datasets were processed and analyzed using DiscoverMP (Refeyn Ltd, Oxford, UK). Details of MP image analysis have been described previously (Sonn-Segev et al, 2020).

## Rubisco inhibition assays

To assess inhibition of Rubisco variants by RuBP, XuBP, or CABP, 20 μM unactivated Rubiscos were incubated with 50 mM HEPES-KOH (pH 8.00), 4 mM EDTA, and either 3 mM RuBP, 1 mM XuBP, or 0.1 mM CABP for 30 min. Subsequently, inhibited Rubiscos were 1:10 diluted into standard photospectrometric assay reactions containing 100 mM HEPES-KOH (pH 8.00), 0.3 mM NADH, 10 mM $MgCl_2$, 2.5 U/mL pgk, 5 U/mL gapdh, 5 mM ATP, 0.02 mg/mL carbonic anhydrase, 70 mM $NaHCO_3$, and 3 mM RuBP. Reaction progress was followed via quantifying the decrease in absorption at 340 nm.

To assess the positive effects XuBP or CABP on activation, activation mixtures (50 mM HEPES-KOH (pH 8.00), 10 mM $MgCl_2$, 20 mM $NaHCO_3$, 0.02 mg/mL carbonic anhydrase, 20 μM Rubisco) were supplemented with 1 mM XuBP or 0.1 mM CABP, prior to 1:10 dilution into the aforementioned assay mixtures.

## Crystallization and structure determination

The sitting-drop vapor-diffusion method was used for crystallization at 16 °C. Purified AncL+7 R269W E271R L273N (10 mg/mL) was incubated with 300 μM CABP and 4.8 mM $MgCl_2$ for 1 h at 3% (v/v) $CO_2$, prior to 1:1 mixing with 100 mM TRIS-HCl (pH 8.5), 200 mM $MgCl_2$, 30% (v/v) polyethylene glycol 400. Crystals appeared within 2 days and were flash frozen in liquid nitrogen.

For the co-crysallization of AncL+7 R269W E271R L273N with AncSSU, purified Rubisco was incubated with a 4-fold molar excess of purified AncSSU at 25 °C for 30 min, prior to purification of the L8S8 complex by size exclusion chromatography (see protein purification section).

Reconstituted L8S8 complex (9 mg/mL) was incubated with 350 μM CABP and 5.6 mM $MgCl_2$ for 1 h at 3% (v/v) $CO_2$, prior to 1:1 mixing with 200 mM BIS-TRIS propane (pH 9.1), and 20% (w/v) polyethylene glycol 4000. Crystals appeared within 2 days. The mother liquor was supplemented with 25% (v/v) PEG200 before crystals were flash frozen in liquid nitrogen.

X-ray diffraction data (Appendix Table S3) were collected at the beamline PETRA III P14 of the DESY (Deutsches Elektronen-Synchrotron, Hamburg). Data were processed with the XDS software package (Kabsch, 2010). Structures were solved by molecular replacement using Phaser of the Phenix software package(Adams et al, 2010) (v.1.1.14), and refined with Phenix.Refine. Additional modeling, manual refinement, and ligand fitting was done in Coot (Emsley and Cowtan, 2004) (v.0.9.8.3). Final positional and B-factor refinements, as well as water picking, were performed using Phenix.Refine. Structural models for the L8 and

L8S8 complex of AncL+7 R269W E271R L273N were deposited to the Protein Data Bank in Europe (PDBe) under PDB accession 8QMV and 8QMW, respectively. Figures were made using PyMOL 2.4.1.

## Hydrogen-deuterium exchange (HDX) mass spectrometry

HDX-MS experiments on Rubisco were carried out as described previously (Osorio-Valeriano et al, 2019) with minor modifications. The conformational dynamics of Rubisco's LSU was studied in 2 different states, i.e., LSU and LSU/CABP (see Source Data HDX). Depending on the investigated state, LSU and SSU were either used individual or mixed to reach final protein concentrations of 25 μM. Where indicated, CABP was present at 50 μM. These batch solutions were incubated in 3% (v/v) $CO_2$ atmosphere for 60 min at 30 °C to facilitate carbamylation of AncL+7 residue 187 (Schulz et al, 2022a). The protein batch solutions were stored in a cooled tray (1 °C) from which 7.5 μL were withdrawn by a robotic autosampler unit (LEAP technologies) and mixed with 67.5 μL of $D_2O$-containing buffer (20 mM Tricine-Na pH 8.0, 75 mM NaCl) to start a HDX reaction. After 10, 30, 95, 1000, or 10,000 s of incubation in another tray at 25 °C, 55 μL samples were taken from the reaction and mixed with an equal volume of pre-dispensed quench buffer (400 mM $KH_2PO_4/H_3PO_4$, 2 M guanidine-HCl, pH 2.2) kept at 1 °C. 95 μl of the resulting mixture were injected through a 50 μl sample loop into an ACQUITY UPLC M-Class System with HDX Technology (Waters) (Wales et al, 2008). Undeuterated samples were generated and treated similar except that $H_2O$-containing buffer was employed for dilution followed by incubation for 10 s. The injected samples were flushed out of the loop with $H_2O$ + 0.1% (v/v) formic acid at 100 μl/min flow rate, guided to an Enzymate BEH Pepsin Column (5 μm, 2.1 × 30 mm (Waters)) containing immobilized porcine pepsin for proteolytic digestion of the proteins at 12 °C, and the resulting peptic peptides collected on an ACQUITY UPLC BEH C18 VanGuard Pre-column (1.7 μm, 2.1 mm × 5 mm (Waters)) kept at 0.5 °C. After 3 min of digestion and trapping, the trap column was placed in line with an ACQUITY UPLC BEH C18 column (1.7 μm, 1.0 × 100 mm (Waters)), and the peptides were eluted at 0.5 °C using a gradient of $H_2O$ + 0.1% (v/v) formic acid (eluent A) and acetonitrile + 0.1% (v/v) formic acid (eluent B) at a flow rate of 30 μl/min as follows: 0–7 min/95–65% A, 7–8 min/65–15% A, 8–10 min/15% A, 10–11 min/5% A, 11–16 min/95% A. The peptides were ionized with an electrospray ionization source (250 °C capillary temperature, 3.0 kV spray voltage), and mass spectra were acquired in positive ion mode over a range of 50 to 2000 $m/z$ on a G2-Si HDMS mass spectrometer with ion mobility separation (Waters) using Enhanced High Definition MS (HDMS[E]) or High Definition MS (HDMS) mode for undeuterated and deuterated samples, respectively (Geromanos et al, 2009). Lock mass correction was implemented with [Glu1]-Fibrinopeptide B standard (Waters). During separation of the peptides, the protease column was washed three times by injecting 80 μl of 0.5 M guanidine-HCl in 4% (v/v) acetonitrile. Additionally, blank injections were performed between each sample to minimize peptide carry-over. All measurements were carried out in three technical replicates.

Peptides were identified from the undeuterated samples (acquired with HDMS[E]) with search parameters as described

previously (Osorio-Valeriano et al, 2019) with the software ProteinLynx Global SERVER (PLGS, Waters), using low energy, elevated energy, and intensity thresholds of 300, 100, and 1000 counts, respectively, and matched using a database containing the amino acid sequences of LSU, SSU, porcine pepsin, and their reversed sequence. After automated processing of the spectra with the software (Waters), all spectra were inspected manually and, if necessary, peptides were omitted (e.g., in case of a low signal-to-noise ratio or the presence of overlapping peptides).

### Experimental study design

No statistical methods were used to estimate sample sizes, and no blinding was done. Sample size was chosen in accordance with the effect differences between experiments. All data was included unless replicates failed during experimental setup. Significance was tested by unpaired, parametric *t*-test, assuming that sample populations have the same SD, which is in line with the experimental setup.

## Data availability

All source data is deposited alongside the publication and made available by the publisher. The atomic structures reported in this paper were deposited to the Protein Data Bank in Europe under accession codes 8QMV and 8QMW.

The source data of this paper are collected in the following database record: biostudies:S-SCDT-10_1038-S44318-024-00311-1.

## Peer review information

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

## Acknowledgements

The X-ray diffraction data was collected at the P14 beamline of the PETRA III storage ring (DESY, Hamburg, Germany). We would like to thank Selina Storm for the assistance at the beamline. We would like to thank O. Mueller-Cajar and Zhijun Guo for help with synthesizing ribulose-1,5-bisphosphate. The authors are grateful for generous support from the Max Planck Society (Max-Planck-Gesellschaft). GKAH is grateful for funding through the EMBO Young Investigator program. LS thanks the Joachim Herz Foundation for support in form of an Add-On fellowship for Interdisciplinary Life Sciences.

## Author contributions

**Luca Schulz**: Conceptualization; Data curation; Formal analysis; Validation; Visualization; Methodology; Writing—original draft; Project administration; Writing—review and editing. **Jan Zarzycki**: Data curation; Software; Formal analysis; Validation; Writing—review and editing. **Wieland Steinchen**: Data curation; Software; Validation; Visualization. **Georg K A Hochberg**: Conceptualization; Resources; Formal analysis; Supervision; Funding acquisition; Validation; Investigation; Methodology; Writing—original draft; Writing—review and editing. **Tobias J Erb**: Conceptualization; Resources; Formal analysis; Supervision; Funding acquisition; Investigation; Writing—original draft; Writing—review and editing.

Source data underlying figure panels in this paper may have individual authorship assigned. Where available, figure panel/source data authorship is listed in the following database record: biostudies:S-SCDT-10_1038-S44318-024-00311-1.

## Funding

## Disclosure and competing interests statement

The authors declare no competing interests.

