## [Peer Review File · The EMBO Journal]

Layered entrenchment maintains essentiality in the evolution of Form I Rubisco complexes.

Georg Hochberg, Tobias Erb, Luca Schulz, Jan Zarzycki, and Wieland Steinchen

Corresponding author(s): Georg Hochberg (georg.hochberg@mpi-marburg.mpg.de) , Tobias Erb (toerb@mpi-marburg.mpg.de),

Review Timeline:

Submission Date:	16th Feb 24
Editorial Decision:	17th Apr 24
Revision Received:	4th Sep 24
Editorial Decision:	18th Oct 24
Revision Received:	23rd Oct 24
Accepted:	28th Oct 24

Editor: Yehu Moran

Transaction Report:

Dear Dr. Erb,

Thank you for submitting your manuscript for consideration by the EMBO Journal. It has now been seen by three referees. Firstly, I would like to apologize for the long time it took us to get back to you with the reviews. Yet, I am happy to inform you that all three Referees found your work interesting and novel as you will see in their reports below.

Given the referees' positive recommendations, I would like to invite you to submit a revised version of the manuscript, addressing the comments of all three reviewers. I should add that it is EMBO Journal policy to allow only a single round of revision, and acceptance of your manuscript will therefore depend on the completeness of your responses in this revised version.

Thank you for the opportunity to consider your work for publication. I look forward to your revision.

Yours sincerely,

Yehu Moran
Academic Editor
The EMBO Journal

We realize that it is difficult to revise to a specific deadline. In the interest of protecting the conceptual advance provided by the work, we recommend a revision within 3 months (16th Jul 2024). Please discuss the revision progress ahead of this time with the editor if you require more time to complete the revisions.

Referee #1:

The authors in this manuscript have built on their previous research of utilizing ancestral reconstruction to investigate the role of Rubisco small subunits in the evolution of Rubisco catalysis. The authors in this paper explored the characteristics that underpin the dependence of the Rubisco octamer on small subunits for assembly and catalysis. The authors took a deep dive into understanding the key amino acids that influence octamer formation and the binding of SSu to alter Rubisco catalytic properties and stabilize the hexadecamer. It has been established in the literature that key loops of the SSu that are peripheral to the central solvent channel impact specificity of Rubisco (see work by Spreitzer PNAS 2005) even though they do not form a part of the active site.

The authors presented an excellent set of experiments to investigate the biochemical components underpinning "persistent essentiality" of the Rubisco small subunits in oligomeric stability and enzyme catalysis. The evidence presented suggested that there could be different mechanisms of entrenchment where for example catalytic neutral residues may exist and the oligomeric structure and assembly of L8 Rubisco core becomes fixed and reliant on SSu's for assembly/stability. The authors initially showed that three mutations added to AncL+7 (Ancestral Rubisco with the ability to bind Rubisco SSu's) REL destabilised the AncL+7 L8 core and in the absence of AncSSu would fall apart into L2 dimers.

It was evident in the results that the binding of AncSSu provided oligomeric stability that was concluded by the authors to provide a stability pathway for new sequence space for Rubisco evolution to occur as possible oligomeric disrupting mutations at the dimer-dimer interface(s) were protected by the SSU's from separation into L2 dimers rendering them inactive. Therefore, new catalytic and structural fitness states of the enzyme could possibly be achieved. This is important as it is well known that a single mutation in Rubisco can cause solubility issues or incompatibility with folding proteins rendering the protein catalytically inactive.

Comments:

- 1) Overall, I thought the manuscript introduction could have written in such a way to attract broader readership. While the concept of essentiality is easily presented, the authors could be more explicit in explaining / introducing the entrenchment concept and the how essentiality of certain protein components can influence this. This needs to be more accessible to readers outside of this field.
- 2) In Figure 2A the relative activity of AncL +7 minus and plus AncSSU were substantially different, however in the previous paper published in Science (Figure 3C), with 2.2 mM RuBP at 25C the relative activities between the L8 and L8S8 were identical. Is there a difference in the Rubisco concentrations used in figure 2A or the assay CO₂ concentration that would account for this difference? The figure legend for Figure 2A should state if the assays were 14CO₂ fixation assays and the assay temperature. Figure 3D also showed differences in relative activity between AncL+7 and AncL+7 + AncSSU.
- 3) In the legend for figure 3G, the Km RUBP assays were coupled enzyme assays and this should also be stated and the assay temperature as well.
- 4) Interestingly, it seems that XuBP provided some cooperativity between oligomer formation and catalysis. The authors conclude that the loose binding of XUBP enables enough structural integrity for catalysis to proceed without AncSSU. Could it be that only one or two sites had XuBP bound and the other active sites were catalytically active?
- 5) Could a molecular crowding agent such as PEG improve octamer formation in AncL+7REL? The high concentration of protein induced octamer formation and therefore could be a molecular crowding effect that is observed.
- 6) I also wonder if the layering of entrenchment involved the numerous factors important for Rubisco assembly. For example, in cyanobacteria it has been shown that rbcX binds specifically to a region of RBCL and stabilises the L8 core prior to SSu addition, whereas in plants it has been shown that Raf 1 (and to an extent RbcX), which has co-evolved with its cognate Rubisco, stabilises the octamer for binding of BSDII (which is promiscuous between plant Rubisco's) and SSu binding displaces BSDII. It could be possible that layering was also dependent on these factors which could have entrenched them for their complex Rubisco folding and assembly pathways. Perhaps, the dependency on Raf1 also enabled additional sequence space to

be accessed as the octamer is stabilised for SSu binding. Reversing the requirement of Raf 1 would be interesting and similarly for RbcX.

Minor Edits:(Because of no line numbers I have quoted the manuscript text and suggested the change)

1) MS: "Entrenchment by such functionally neutral mutations can persist if entrenching mutations are particularly likely to occur and unlikely to revert because of biases in the mutational process".
In the introduction the authors focused on essentiality and then moved to Entrenchment which requires more explicit introduction and explanation.

2) MS: "In the case of Rubisco this explanation seems unlikely: While the initial recruitment of the SSU was probably catalytically beneficial through an allosteric effect on Rubisco's active site, substitutions that subsequently made the interaction essential, such that Rubisco could no longer function without it, apparently did not improve the enzyme at all".
From 'apparently', this statement is confusing as plant Rubisco has superior kinetics in air compared to the bacterial enzymes - The carboxylation efficiency of these enzymes is also superior compared to AncL and AncLS.

3) MS:"The three substitutions are localized in proximity to the Rubisco's dimer-dimer interface"
Please be more specific which residues you are referring to - REL.

4) Relative activity - This term was used for the activity assays - can authors state which is this relative too? Is it AncL+7? In some figure legends this is not stated.

Referee #2:

This paper examines how a protein complex with enzymatic activity has evolved essential subunits. The paper combines phylogenetic reconstruction with enzymatic assays and structural biology analyses. The questions addressed are important and have broad impacts on our understanding of evolution at the biochemical level. Indeed, we know that genes, their products, and their interactions become essential over time across cellular functions, but how this essentiality builds up has not been resolved in most cases. This work offers a very powerful system to ask this question with a very important enzyme.

General comments:

I am not a specialist in some of the experimental methods used, so I cannot comment on them.

The paper is generally written in a very dense and specialized language. This makes it nicely short but difficult to read for non-specialists or experts in the field unfamiliar with the group's previous papers on the same topic. If the journal allows it, I recommend using more space to explain concepts and the necessary background information better, as well as the experimental results.

The abstract states that Rubisco rapidly depended on a newly recruited interaction partner. How do we know this was rapid if we cannot say which mutations occurred and when? A few mutations suggest that it could have happened quickly, but without actual dates, we cannot say so. The issue of temporality and how many mutations were needed for something to occur is often based on the paper's implicit assumptions. It would be good to make those assumptions explicit. Unless I am missing something, from the three presented in Figure 1, it is impossible to establish the order of establishment of mutations, so conclusions based on temporal events are based on assumptions that can be justified but not tested, so it would be important to make them explicit.

It is unclear if the authors test the question they elaborate on in the introduction. It seems that they want to know >why< the dependency on SSU has become essential, but since they cannot determine in which temporal order the mutations that have occurred since ANCL, and that epistasis could also be important among all of the mutations that have occurred since, how do we know which changes or combination of changes mattered historically?

Much of Figure 1 is from their previous paper, so it would be helpful for the authors to assume the reader has not read it and present the data accordingly. Overall, it is excellent, but maybe more text around it would help.

This sentence here < This is at odds with the observation that once a component becomes essential, it tends to stay essential, as is the case for Rubisco's SSU12-15.> suggests that the non-reversibility of essentiality is a general phenomenon, but the citations only concern Rubisco. Is this a general observation? Some recent studies using experimentally evolved microbes show that genes can frequently become essential and dispensable, in both directions. For instance, see: <https://www.science.org/stoken/author-tokens/ST-1664/full>

Flipping essentiality is also observed in protein complexes: <https://academic.oup.com/gbe/article/5/6/1049/616770>

It would be important to specify here if the biases in the mutational process refer to the actual changes in sequences or are biased in terms of effects > Entrenchment by such functionally neutral mutations can persist if entrenching mutations are particularly likely to occur and unlikely to revert because of biases in the mutational process>

The first parts of the results were well-developed, but the second part was more dense and summarized. Expanding the text to help walk us through the results would be good.

It is not clear why ancestral SSU is used, and most importantly, it is not specified which one is used in the text and figures, as the authors sometimes refer to SSU in the figures but AncSSu in the text.

The HDX experiment reveals sites already known to be involved in catalysis (the authors cite refs 24 and 25). I think it is acceptable for the authors to include this data here, but it is unclear how critical it was to find sites to investigate. It could be better to mention what we knew before the HDX experiments in the text and why it was not enough to find sites to investigate.

The amount of SSU used in experiments appears to change (5-fold excess in Figure 3D and 10-fold excess in Figure 3F). The need to change this condition and the consequences for the conclusions from these experiments need to be explained and justified. The impacts appear dramatic, as in all other experiments, mutants with SSU never reach levels of AncL+7, whereas in 3D, they overshoot it. This section is an example of a section where the results are too succinct to follow the reasoning and experiments very well.

In the discussion, explaining the link between mutational biases would be helpful. Also, again discuss how we can conclude that something needed only a few changes to occur if we do not know the order in which mutations occurred.

Minor comments

The second sentence of the main text: Should the "components" at the beginning be singular rather than plural to avoid confusion regarding the essentiality of multiple components versus the essentiality of individual components?

References 6 and 7 are the same, and many references are incomplete.

In the introduction, papers regarding subfunctionalization are cited, but gene duplication is not mentioned. It could have been good to mention duplication since this is one of the major ways by which protein complexes gain subunits that later become essential. < Such complexes develop when a novel subunit is recruited into a simpler ancestral complex, which subsequently evolves to completely depend on the new component3-6>

The manuscript did not have line or page numbers, which made it difficult to review and comment on.

Referee #3:

In this work Schulz and colleagues delve deeper into the evolution of function and form of Form I rubisco. They perform a detailed and complex set of experiments unpick molecular dependencies between the two subunits that influence the formation of the multimer, its kinetics and its solubility. The work is excellent, detailed, and innovative. I applaud the authors efforts and ingenuity here. I only have minor suggestions that are detailed below. These mostly concern the presentation of the data and the extent of the claims made. All of these can be addressed by minor text changes or provisional of statistical testing results.

General

- The title is too general given the subject of the work on one protein complex. It really should focus on the dependency between RbcL and RbcS. Overclaiming generality is does more harm than good.
- The authors sometimes use entrenchment quite loosely. It appears to mean "dependency" of some property of the enzyme on a single [or small set] amino acid substitution. Sometimes this dependency is solubility [filament formation], sometimes its kinetics, sometimes if multimer formation. It would be good to be clearer about this throughout the manuscript to help the reader understand the key messages of the work. Referring to all dependencies as "entrenchment" doesn't quite capture the results that are presented.
- Some claims are too absolute. For example, many of the results indicate potential mechanisms but do not prove them or exclude other mechanisms. It would be good to acknowledge the alternative explanations that the authors feel are less likely, but that are not necessarily ruled out by the analyses presented here.

Abstract

- I'm not sure that the first sentence of the abstract is strictly true - the claim about "gradual complexification" and is false. Its just one way in which the process can occur.
- Essentiality can arise during co-evolution... there are other ways for it to arise.
- In general I think the "absoluteness" of the claims need to be moderated.

Introduction

- Which cannot function if any of their components are missing > whose function is perturbed (the reason being is that several multiprotein complexes have severely reduced activity in the absence of some subunits as is shown in several of the graphs in this work).
- This process provides an explanation of how seemingly irreducible biological complexity could have evolved => this is a bit over the top. It helps explain why the monomeric versions of proteins that form multi-subunit complexes appear to be absent from extant sequenced genomes.
- Again in general, I would encourage the authors to be less absolute and portray a more realistic and nuanced review of the data rather than the binary yes/no view that is presented.

Results

- It would be helpful to define the various AncL, AncLS and AncL+7 clearly at the start of the results. Getting a clear picture of these is essential to be able to understand the results and its hard to unpack this from the introduction and from the information provided at the start of the results.
- In general I found all of the results to be clearly presented and supported by extensive supplementary material.
- There was a general absence of statistical testing when reporting whether things were different or not. In most cases the differences are quite large. But in some cases the differences don't appear to be substantial but are claimed to be so. Support via appropriate statistical testing would help support these claims. Particularly where large variation appears to be present between what are experimental replicates (e.g. Figure 3D and F have the same enzymes (AncL+7) under the same conditions but have very different activity values +/- the SSU (75% in panel D and 50% in panel F).

Discussion

- "Mutation bias" is not the right term here.
- It would benefit from a summary table or figure or some way to synthesise the large number of enzyme variants interrogated and discussed in this work.

See also guidelines for figure

legends: <https://www.embopress.org/page/journal/14602075/authorguide#figureformat>

Further information is available in our Guide For

Authors: <https://www.embopress.org/page/journal/14602075/authorguide>

We realize that it is difficult to revise to a specific deadline. In the interest of protecting the conceptual advance provided by the work, we recommend a revision within 3 months (16th Jul 2024). Please discuss the revision progress ahead of this time with the editor if you require more time to complete the revisions. Use the link below to submit your revision:

EMBO Journal Revisions

Please note that when checking and re-validating the source data of all graphs, we noticed that the data in Figure 3C was accidentally normalized to the turnover of a wrong variant (0.26/s instead of 0.28/s). We have corrected this mistake in the revised Figure 3C. The overall difference is marginal and no conclusions change. The correct source data is attached.

We changed Figure 1C to now contain arrows stating "CO₂ → metabolism" rather than "CO₂ → sugar" to make the intended meaning of these schemes clearer.

We adjusted the color scheme throughout the figures to be consistent with Figure 1D.

Referee #1:

The authors in this manuscript have built on their previous research of utilizing ancestral reconstruction to investigate the role of Rubisco small subunits in the evolution of Rubisco catalysis. The authors in this paper explored the characteristics that underpin the dependence of the Rubisco octamer on small subunits for assembly and catalysis. The authors took a deep dive into understanding the key amino acids that influence octamer formation and the binding of SSu to alter Rubisco catalytic properties and stabilize the hexadecamer. It has been established in the literature that key loops of the SSu that are peripheral to the central solvent channel impact specificity of Rubisco (see work by Spreitzer PNAS 2005) even though they do not form a part of the active site.

The authors presented an excellent set of experiments to investigate the biochemical components underpinning "persistent essentiality" of the Rubisco small subunits in oligomeric stability and enzyme catalysis. The evidence presented suggested that there could be different mechanisms of entrenchment where for example catalytic neutral residues may exist and the oligomeric structure and assembly of L8 Rubisco core becomes fixed and reliant on SSu's for assembly/stability. The authors initially showed that three mutations added to AnCL+7 (Ancestral Rubisco with the ability to bind Rubisco SSu's) REL destabilised the AnCL+7 L8 core and in the absence of AncSSu would fall apart into L2 dimers.

It was evident in the results that the binding of AncSSUE provided oligomeric stability that was concluded by the authors to provide a stability pathway for new sequence space for Rubisco evolution to occur as possible oligomeric disrupting mutations at the dimer-dimer interface(s) were protected by the SSU's from separation into L2 dimers rendering them inactive. Therefore, new catalytic and structural fitness states of the enzyme could possibly be achieved. This is important as it is well known that a single mutation in Rubisco can cause solubility issues or incompatibility with folding proteins rendering the protein catalytically inactive.

We thank the reviewer for the positive assessment of our work and for the in depth comments that helped improve the manuscript. We now additionally make sure to cite the mentioned Spreitzer PNAS 2005 paper when discussing sites that were previously known / implicated to be involved in catalysis.

Comments:

1) Overall, I thought the manuscript introduction could have written in such a way to attract

broader readership. While the concept of essentiality is easily presented, the authors could be more explicit in explaining / introducing the entrenchment concept and the how essentiality of certain protein components can influence this. This needs to be more accessible to readers outside of this field.

We agree with the reviewer and have thus taken measures to help attract a broader readership to the article. This includes a more detailed introduction to the concept of entrenchment, a more in depth introduction into the protein variants that were studied (and their origin), and an overview table of studied variants and their phenotypes of interest (Appendix Table S1 and S2).

2) In Figure 2A the relative activity of AncL +7 minus and plus AncSSU were substantially different, however in the previous paper published in Science (Figure 3C), with 2.2 mM RuBP at 25C the relative activities between the L8 and L8S8 were identical. Is there a difference in the Rubisco concentrations used in figure 2A or the assay CO₂ concentration that would account for this difference? The figure legend for Figure 2A should state if the assays were ¹⁴C₂ fixation assays and the assay temperature. Figure 3D also showed differences in relative activity between AncL+7 and AncL+7 + AncSSU.

We added a “Measurement of relative Rubisco activities” section to the materials and methods. The assays in question are identical to the ones that we used to assess Rubisco inhibition, which we describe at a later stage of the Material and Methods. However, we apparently failed to introduce them earlier in the context of measuring relative Rubisco activities. We thank the reviewer for drawing our attention to this mistake. We also adjusted the legends of Figure 2A, D and Figure 3C, D, and F and now describe the assays more accurately and refer to the correct paragraphs in the methods section.

Regarding the differences between the two datasets mentioned by the reviewer:

Data for the referenced Figure 3C of the Science paper was collected using a different assay (see “Rubisco catalysis at elevated temperatures” in the Supplementary Materials of the referenced publication). While there are multiple differences in the assay setup, the most important difference is that in Figure 3C of the previous publication, data was collected at a lower NaHCO₃ concentration (45 mM as opposed to 70 mM). This was done to ensure that the CO₂ concentration in solution could be kept constant throughout the assay at higher temperatures (where much more NaHCO₃ is needed to reach the same concentration of dissolved CO₂). Note that the K_m(CO₂) of AncL+7 decreases ~4-fold upon AncSSU addition, so that in the earlier assays with 45 mM NaHCO₃, the observed rates are similar, because AncL+7 is not fully saturated under these conditions (K_m(CO₂) ~500 μM), whereas AncL+7 + AncSSU is already saturated (K_m(CO₂) ~150 μM). At CO₂ saturation (70 mM NaHCO₃), however the decrease in relative activity between AncL+7 with and without addition of AncSSU is expected and fully in line with the Michaelis Menten data we published previously. At saturation, the k_{cat} of AncL+7 decreases from 0.63 to 0.41 upon AncSSU addition (see Table 1 of the previous publication). The decrease in relative activity to ~60-65% upon AncSSU addition shown in Figure 2A, 3D, etc. is thus in line with previous data.

3) In the legend for figure 3G, the K_m RUBP assays were coupled enzyme assays and this should also be stated and the assay temperature as well.

We added the following statement to the figure legend: “Catalysis rate at 25 °C was determined by following the rate of NADH oxidation using a coupled enzyme assay described in detail in the methods section”. We now also list the assay temperature in the relevant methods section.

4) Interestingly, it seems that XuBP provided some cooperativity between oligomer formation and catalysis. The authors conclude that the loose binding of XuBP enables enough structural integrity for catalysis to proceed without AncSSU. Could it be that only one or two sites had XuBP bound and the other active sites were catalytically active?

We considered this as a possibility but deem it unlikely because we did not see indications of persistently bound XuBP to Rubisco. This is based on two observations: 1) we did not observe a loss in turnover when XuBP was present in the assay of octameric variants and 2) the presence of XuBP in the activation mixture (where RuBP is absent) did not inhibit Rubisco activity (Extended Data Figure 5A). We reasoned that under these activation conditions Rubisco variants should be saturated with XuBP until they come in contact with RuBP of the assay mixture. If XuBP would only slowly release from the Rubisco octamer (or stayed bound persistently), we would expect to see a slower initial rate of NADH oxidation in the presence of XuBP.

To make it clearer that XuBP was present during both the activation and in the final assay, we adjusted the figure legend of relevant figures (e.g., Extended Data Figure 5A).

We now state that “Notably, no decrease in catalytic rate was observed when 1 mM XuBP was added to both the activation and assay mixture (Appendix Figure S5).” prior to drawing the conclusion that XuBP is likely not tightly bound to the active site of the investigated Rubisco variants.

5) Could a molecular crowding agent such as PEG improve octamer formation in AncL+7REL? The high concentration of protein induced octamer formation and therefore could be a molecular crowding effect that is observed.

This is a possibility, but we also expect this effect purely from the thermodynamics of subunit-subunit binding (higher concentration leading to more binding). A self-crowding effect seems less likely to us, because self-crowding-induced oligomerization would lead to a substantial decrease in crowding when most subunits begin to associate into octamers (there would now be $\frac{1}{4}$ the number of particles). We ultimately chose not to include PEG experiments because of time constraints, and because our main point here was to show that oligomerization induced by something other than XuBP has the same effect.

6) I also wonder if the layering of entrenchment involved the numerous factors important for Rubisco assembly. For example, in cyanobacteria it has been shown that rbcX binds specifically to a region of RBCL and stabilises the L8 core prior to SSu addition, whereas in plants it has been shown that Raf 1 (and to an extent RbcX), which has co-evolved with its cognate Rubisco, stabilises the octamer for binding of BSDII (which is promiscuous between plant Rubisco's) and SSu binding displaces BSDII. It could be possible that layering was also dependent on these factors which could have entrenched them for their complex Rubisco folding and assembly pathways. Perhaps, the dependency on Raf1 also enabled additional

sequence space to be accessed as the octamer is stabilised for SSu binding. Reversing the requirement of Raf 1 would be interesting and similarly for RbcX.

This is of course an interesting thought and work is well underway in our groups to answer this question.

Minor Edits:(Because of no line numbers I have quoted the manuscript text and suggested the change)

We apologize for the lack of line numbers and have now added them to support the revision process.

1) MS: "Entrenchment by such functionally neutral mutations can persist if entrenching mutations are particularly likely to occur and unlikely to revert because of biases in the mutational process".

In the introduction the authors focused on essentiality and then moved to Entrenchment which requires more explicit introduction and explanation.

We now more explicitly introduce and explain entrenchment in the introduction.

2) MS: "In the case of Rubisco this explanation seems unlikely: While the initial recruitment of the SSU was probably catalytically beneficial through an allosteric effect on Rubisco's active site, substitutions that subsequently made the interaction essential, such that Rubisco could no longer function without it, apparently did not improve the enzyme at all". From 'apparently', this statement is confusing as plant Rubisco has superior kinetics in air compared to the bacterial enzymes - The carboxylation efficiency of these enzymes is also superior compared to AnCL and AnCLS.

We agree with the reviewer that Rubisco certainly became more efficient over the course of evolution after the SSU had evolved. This is unrelated to our statement: We meant that the substitutions that initially made the interaction essential did not improve function. This is described in our previous paper. Of course, subsequently plant lineage Form I Rubiscos evolved improved kinetics.

We have re-phrased the statement in question to: "First, the substitutions that caused Rubisco to become dependent on the SSU for solubility do not appear to have improved Rubisco's performance when they first arose. It is therefore unlikely that the SSU's essentiality is maintained as a secondary effect of purifying selection acting on a functional benefit associated with the entrenching substitutions."

This is not definitive statement to say that all subsequent substitutions that had an influence on the essentiality of the LSU-SSU interaction across all of history were necessarily functionally neutral. It only means that the initial entrenchment appears to have been neutral.

3) MS:"The three substitutions are localized in proximity to the Rubisco's dimer-dimer interface"

Please be more specific which residues you are referring to - REL.

We agree that the text was ambiguous and thank the reviewer for pointing this out. We have adjusted the text to be more specific by listing/better describing the relevant substitutions throughout the paragraph.

4) Relative activity - This term was used for the activity assays - can authors state which is this relative too? Is it AnCL+7? In some figure legends this is not stated.

We have adjusted all relevant figure legends to state which measurements the activities are relative to.

Referee #2:

This paper examines how a protein complex with enzymatic activity has evolved essential subunits. The paper combines phylogenetic reconstruction with enzymatic assays and structural biology analyses. The questions addressed are important and have broad impacts on our understanding of evolution at the biochemical level. Indeed, we know that genes, their products, and their interactions become essential over time across cellular functions, but how this essentiality builds up has not been resolved in most cases. This work offers a very powerful system to ask this question with a very important enzyme.

We thank the reviewer for the positive assessment and for highlighting the impact and importance of our work.

General comments:

I am not a specialist in some of the experimental methods used, so I cannot comment on them.

The paper is generally written in a very dense and specialized language. This makes it nicely short but difficult to read for non-specialists or experts in the field unfamiliar with the group's previous papers on the same topic. If the journal allows it, I recommend using more space to explain concepts and the necessary background information better, as well as the experimental results.

We agree with the Reviewer's comments (and those of other Reviewers). We now use more space to explain concepts and the necessary background information better. This includes a more comprehensive introduction into the concept of entrenchment (introduction), an introductory paragraph to the results, and generally a more descriptive results section.

The abstract states that Rubisco rapidly depended on a newly recruited interaction partner. How do we know this was rapid if we cannot say which mutations occurred and when? A few mutations suggest that it could have happened quickly, but without actual dates, we cannot say so. The issue of temporality and how many mutations were needed for something to occur is often based on the paper's implicit assumptions. It would be good to make those assumptions explicit. Unless I am missing something, from the three presented in Figure 1, it is impossible to establish the order of establishment of mutations, so

conclusions based on temporal events are based on assumptions that can be justified but not tested, so it would be important to make them explicit.

We agree with the reviewer that we do not have knowledge of timescales or temporal order of events over the interrogated & presented evolutionary interval. We have re-phrased the abstract and introduction to now talk about “easily accessible”, rather than “rapid” dependence, which is more reflective of the intended meaning of our statement and removes the temporal element.

The lack of temporal order was already part of the discussion in our initial submission but we agree that this is integral to the expectation set throughout the paper.

It is unclear if the authors test the question they elaborate on in the introduction. It seems that they want to know >why< the dependency on SSU has become essential, but since they cannot determine in which temporal order the mutations that have occurred since ANCL, and that epistasis could also be important among all of the mutations that have occurred since, how do we know which changes or combination of changes mattered historically?

We believe that the changes made to the introduction (see above) now better set the expectation going into the study. We highlight proof-of-principle cases in which single (or low numbers of) substitutions can cause the dependence on new interaction partner through diverse biochemical mechanisms. Similarly, we revert the substitutions that were identified in the “forward direction” (creating a dependence) and show that this is not sufficient to alleviate dependencies. This implies that there must be more entrenching states, epistatic interactions, and likely additional mechanisms that remain to be discovered (as discussed in the the discussion).

What makes us confident that these changes, or a combination thereof, mattered historically is the fact that even in ~100 amino acid derived / more evolved ancestral proteins (e.g., AncLS), we can partially or fully alleviate dependencies by reverting causal substitutions identified from our “forward” proof-of-principle assays.

Much of Figure 1 is from their previous paper, so it would be helpful for the authors to assume the reader has not read it and present the data accordingly. Overall, it is excellent, but maybe more text around it would help.

We thank the Reviewer for the kind words about Figure 1.

In line with the reviewer’s suggestion, we now describe the lead up work/Figure 1 in more detail. This expanded description goes hand-in-hand with a newly introduced “introductory paragraph” to the results section, which should also help clarify the experimental setup to a broader audience.

This sentence here < This is at odds with the observation that once a component becomes essential, it tends to stay essential, as is the case for Rubisco's SSU12-15.> suggests that the non-reversibility of essentiality is a general phenomenon, but the citations only concern Rubisco. Is this a general observation? Some recent studies using experimentally evolved microbes show that genes can frequently become essential and

dispensable, in both directions. For instance, see: <https://www.science.org/stoken/author-tokens/ST-1664/full>

Flipping essentiality is also observed in protein complexes: <https://academic.oup.com/gbe/article/5/6/1049/616770>

Our statement was not precise enough. The reviewer is right that essentiality can be reversed. We have adjusted the text to highlight that it can sometimes be very long lasting (as is the case in the SSU). What mechanisms lead to reversals of essentiality is a very interesting problem, but they are beyond the scope of our study.

It would be important to specify here if the biases in the mutational process refer to the actual changes in sequences or are biased in terms of effects > Entrenchment by such functionally neutral mutations can persist if entrenching mutations are particularly likely to occur and unlikely to revert because of biases in the mutational process>

We rephrased the statement in question to read “In these cases, essentiality can persist only if the likelihood of entrenching substitutions significantly exceeds the likelihood of entrenchment-reversing substitutions” and added a relevant citation. This should be more specific and better ties into the re-worked, longer introduction about entrenchment.

The first parts of the results were well-developed, but the second part was more dense and summarized. Expanding the text to help walk us through the results would be good.

We agree with the Reviewer’s assessment and have expanded the text for both parts of the results, with a particular focus on the second result part of the results.

It is not clear why ancestral SSU is used, and most importantly, it is not specified which one is used in the text and figures, as the authors sometimes refer to SSU in the figures but AncSSu in the text.

We now spend more time explaining our variant selection (introductory paragraph to the results section), have added a variant overview that describes the relevant phenotypes/characteristics of each utilized protein (Appendix Table S1 and S2), and have adjusted text and figures to uniformly refer to “the SSU” as AncSSU. In the initial submission “the SSU” and AncSSU were, at times, used interchangeably, which is ambiguous and has been corrected.

The HDX experiment reveals sites already known to be involved in catalysis (the authors cite refs 24 and 25). I think it is acceptable for the authors to include this data here, but it is unclear how critical it was to find sites to investigate. It could be better to mention what we knew before the HDX experiments in the text and why it was not enough to find sites to investigate.

While the HDX experiment does recover sites that were known to be involved in catalysis, this analysis was absolutely crucial to find sites in regions that we would not have investigated otherwise. In fact, we failed to rationally identify sites with the desired effects on catalysis. We would never have rationally targeted changes such as a leucine to isoleucine

substitution (L192I – shift of 1 methyl group!), had it not been identified to be in an active site-responsive region of the protein.

In justifying which variants we investigated, we now say that “We chose to investigate the effect of ten substitutions in regions with the most drastic HDX response to inhibitor binding, many of which were outside of regions that were previously connected to catalysis in Rubisco.”

The amount of SSU used in experiments appears to change (5-fold excess in Figure 3D and 10-fold excess in Figure 3F). The need to change this condition and the consequences for the conclusions from these experiments need to be explained and justified. The impacts appear dramatic, as in all other experiments, mutants with SSU never reach levels of AncL+7, whereas in 3D, they overshoot it. This section is an example of a section where the results are too succinct to follow the reasoning and experiments very well.

The use of a 10-fold AncSSU excess is a technicality that derives from a decreased affinity of the “fiber interface reversion” construct used in Figure 3F (and 1E, when interrogating the fiber interface reversion construct). This construct was created by reverting sites in the fiber-forming interface back to their ancestral states in AncL (which has no affinity for AncSSU). Because the fiber-forming interface and the AncSSU-binding interface partially overlap (see Schulz et al. 2022, Science, Supplementary Figure S12H) the affinity of the FIR construct towards AncSSU is lower, which was circumvented by employing a 10-fold AncSSU excess to ensure SSU saturation, as was shown previously (see aforementioned publication).

The fact that a 10-fold AncSSU excess was used is likely not the reason for the observed catalytic behavior (see e.g., the catalytic activity of AncL+7, which behaves as expected).

We now comment on the high catalytic rates of the fiber interface reversion construct when it is first introduced (Figure 1E) by stating that: “Additionally, the FIR construct is more active than ancestral variants with more historical substitutions introduced (e.g., 1.58 s⁻¹ for the FIR construct (see Appendix Table S1), versus 0.29 s⁻¹ for AncLS), which highlights the context dependence and epistatic interplay of historical substitutions.”

In the discussion, explaining the link between mutational biases would be helpful. Also, again discuss how we can conclude that something needed only a few changes to occur if we do not know the order in which mutations occurred.

We now frame the mutational bias/mutational propensity more clearly as a previously identified mechanism that can help maintain essentiality (which has now more clearly been introduced as such in the introduction).

The lack of temporal order through which the substitutions occurred is already brought up as a discussion point. We think that together with the re-worked introduction, which now specifically states that no temporal order of substitution acquisition can be established, this paints an accurate picture of our motivations and how we could draw our conclusions.

Minor comments

The second sentence of the main text: Should the “components” at the beginning be singular rather than plural to avoid confusion regarding the essentiality of multiple components versus the essentiality of individual components?

We agree that this helps avoid confusion and have adjusted the statement accordingly.

References 6 and 7 are the same, and many references are incomplete.

We have removed repeat references and made sure that all references are complete. We thank the reviewer for pointing this out.

In the introduction, papers regarding subfunctionalization are cited, but gene duplication is not mentioned. It could have been good to mention duplication since this is one of the major ways by which protein complexes gain subunits that later become essential. < Such complexes develop when a novel subunit is recruited into a simpler ancestral complex, which subsequently evolves to completely depend on the new component3-6>

Good point, we have added duplications as a mechanism to generate new components.

The manuscript did not have line or page numbers, which made it difficult to review and comment on.

We apologize for this. The revised document now features line & page numbers to assist the reviewers in the revision process.

Referee #3:

In this work Schulz and colleagues delve deeper into the evolution of function and form of Form I rubisco. They perform a detailed and complex set of experiments unpick molecular dependencies between the two subunits that influence the formation of the multimer, its kinetics and its solubility. The work is excellent, detailed, and innovative. I applaud the authors efforts and ingenuity here. I only have minor suggestions that are detailed below. These mostly concern the presentation of the data and the extent of the claims made. All of these can be addressed by minor text changes or provisional of statistical testing results.

We thank the reviewer for the highly positive assessment of our work, calling it excellent, detailed, and innovative.

General

- The title is too general given the subject of the work on one protein complex. It really should focus on the dependency between RbcL and RbcS. Overclaiming generality is does more harm than good.

This is a fair point. We have adjusted the title to: "Layered entrenchment maintains essentiality in the evolution of Form I Rubisco complexes"

- The authors sometimes use entrenchment quite loosely. It appears to mean "dependency" of some property of the enzyme on a single [or small set] amino acid substitution. Sometimes this dependency is solubility [filament formation], sometimes its kinetics, sometimes if multimer formation. It would be good to be clearer about this throughout the

manuscript to help the reader understand the key messages of the work. Referring to all dependencies as "entrenchment" doesn't quite capture the results that are presented.

Entrenchment is a fixed (and general) term in the literature that we use in its broadest term to refer to the dependence of (two) protein subunits on each other during evolution (without necessarily knowing the mechanistic context of how these interactions become conserved). In our work, we want to make the point that entrenchment is not based on a single mechanisms, but rather that individual mechanisms are at work, that act together (and independent of each other) at the same time. Thus, we chose to first refer to the general term in the manuscript, and then add descriptors that provide a mechanistic explanation for the individual mechanisms at individual sites. We hope that in the revised version this point comes better across.

- Some claims are too absolute. For example, many of the results indicate potential mechanisms but do not prove them or exclude other mechanisms. It would be good to acknowledge the alternative explanations that the authors feel are less likely, but that are not necessarily ruled out by the analyses presented here.

We went through the manuscript and toned down absolute claims by further elaborating on alternative explanations and adjusting our language.

Abstract

- I'm not sure that the first sentence of the abstract is strictly true - the claim about "gradual complexification" and is false. Its just one way in which the process can occur.

We agree with the reviewer that claiming gradual complexification as the only mechanism by which obligate protein complexes arise is false. We have adjusted the statement to say that they "... often arise ..." through gradual complexification. We think this is a fair statement, as gradual complexification has been shown to have underlie the development of many obligate protein complexes and more complex biological systems (e.g., <https://doi.org/10.1038/nature06942>, [10.1146/annurev-biochem-060614-034142](https://doi.org/10.1146/annurev-biochem-060614-034142), <https://doi.org/10.1038/s41467-019-13026-3>, or <https://doi.org/10.1017/S0952523813000035>).

- Essentiality can arise during co-evolution... there are other ways for it to arise.

We agree and have adjusted the statement to "Essentiality can arise..."

- In general I think the "absoluteness" of the claims need to be moderated.

In line with the reviewer's general comments, we went through the manuscript to moderate the absoluteness of claims and elaborate on alternatives.

Introduction

- Which cannot function if any of their components are missing > whose function is perturbed (the reason being is that several multiprotein complexes have severely reduced activity in the absence of some subunits as is shown in several of the graphs in this work).

We agree that perturbed functions are plausible and can clearly be observed from our data. We have thus adjusted the statement to read “The novel component is now ‘entrenched’, which means that the complex can no longer carry out its full function in the absence of the new subunit.” to signify that “full functionality” is compromised.

- This process provides an explanation of how seemingly irreducible biological complexity could have evolved => this is a bit over the top. It helps explain why the monomeric versions of proteins that form multi-subunit complexes appear to be absent from extant sequenced genomes.

We have deleted this sentence.

- Again in general, I would encourage the authors to be less absolute and portray a more realistic and nuanced review of the data rather than the binary yes/no view that is presented.

We went through the manuscript and toned down the absoluteness of claims by further elaborating on alternative explanations.

Results

- It would be helpful to define the various AnCL, AnCLS and AnCL+7 clearly at the start of the results. Getting a clear picture of these is essential to be able to understand the results and its hard to unpack this from the introduction and from the information provided at the start of the results.

We agree with the reviewer (a similar point was brought up by Reviewer #2). In an attempt to better introduce previous work and help the reader understand the various variants that were used, we now start the results section with an introductory paragraph. Notably, this introductory paragraph also contains a reference to a new Appendix Table S2, which contains an overview and description of all variants used in the presented work.

- In general I found all of the results to be clearly presented and supported by extensive supplementary material.

We thank the reviewer for this assessment.

- There was a general absence of statistical testing when reporting whether things were different or not. In most cases the differences are quite large. But in some cases the differences don't appear to be substantial but are claimed to be so. Support via appropriate statistical testing would help support these claims. Particularly where large variation appears to be present between what are experimental replicates (e.g. Figure 3D and F have the same enzymes (AnCL+7) under the same conditions but have very different activity values +/- the SSU (75% in panel D and 50% in panel F).

We added statistical testing as well as more quantitative numbers for Figure 3D and 3F and now discuss these metrics in more detail in the results section. We additionally added a more exhaustive explanation and discussion of the results presented in Figure 3F.

The reviewer rightfully points out variations in activity values +/- SSU between panel D and F of figure 3, however, the variation is less drastic than mentioned (57% vs 67% as opposed to 50% vs 75%) and they likely stem from day-to-day variations in assay set up / reagents. We consider these variation to be of little influence here, as we only draw conclusions from head-to-head comparisons between self-consistent data (e.g., Figure 3D all measured using the same batch assay setup on the same day and Figure 3F all measured using the same batch setup on another day).

Discussion

- "Mutation bias" is not the right term here.

Mutational bias is an accepted term that describes that not all mutations occur with equal likelihood. It is relevant for entrenchment in that for example the mutations towards hydrophobic amino acids occur with a frequency that makes it unlikely for a hydrophobic surface to become hydrophilic by chance. But we understand that these definitions can be contested and that the paper doesn't need them. We have substituted the term with 'propensity'.

- It would benefit from a summary table or figure or some way to synthesise the large number of enzyme variants interrogated and discussed in this work.

We have added a summary table of all enzyme variants, their phenotypes, and their substitutions, to the supplementary information and refer to it in the introductory paragraph of the results section. The table is listed as "Appendix Table S2".

Dear Dr. Hochberg,

Thank you for submitting your manuscript for consideration by the EMBO Journal. It has now been seen by two of the three original referees whose comments are enclosed.

Given the referees' positive recommendations, I would like to invite you to submit a revised version of the manuscript, addressing the remaining technical comments raised by our editorial assistants (please see below).

We generally allow three months as standard revision time. Yet, I find it highly unlikely you would need such a long time for addressing the few remaining issues.

Thank you for the opportunity to consider your work for publication. I look forward to your revision.

Yours sincerely,
Yehu Moran
Academic Editor
The EMBO Journal

Specific issues raised by the editorial assistants:

- *Keywords: missing, please complete.
- *ORCID ID: Was missing for Dr. Erb. A request was sent by email with instructions how to link his ORCID to his account. Please make sure this is done.
- *AC: remove from manuscript
- *Disclosure and Competing Interests Statement: Please rename.
- *REFERENCE FORMAT: Incorrect. Please use our standard format.
- *APPENDIX 1 FILE WITH ToC: All the appendix data should be removed from the manuscript and uploaded in a separate PDF labelled "Appendix". The appendix needs a table of contents with page numbers.
- *REAGENTS AND TOOLS TABLE: Missing. Please upload it as a separate file using the template we provide in the author guidelines.
- *SYNOPSIS IMAGE: Not provided. Please provide.
- *SYNOPSIS TEXT: Not provided. Please provide.
- *FIGURE CALLOUTS: There are callouts for an Extended Figure 1 and 5, please correct.

Further Notes:

- Please remove the figures from the manuscript and compile the figure legends at the end of the manuscript text.
- The heading "References and Notes" should be deleted.
- Please correct the order of the manuscript sections to: Abstract, Keywords, Introduction, Results, Discussion, Methods, Data Availability, Acknowledgements, Disclosure and Competing Interests Statement, References, Figure legends

DATA CHECK: FAIL. Please check and correct.

- DAS:

1. Please note that the specific URLs for 8QMV and 8QMW datasets are not provided in the data availability statement.
2. Please note that reviewer access codes for 8QMV and 8QMW datasets are not provided in the data availability statement.

- Figure legends:

Please note that the exact p values are not provided in the legends of figures 3d, f. Please correct.

General instructions for preparing your revised manuscript:

- a point-by-point response to the referees' comments, with a detailed description of the changes made (as a word file).
- a word file of the manuscript text.
- individual production quality figure files (one file per figure)

- a complete author checklist, which you can download from our author guidelines (<https://www.embopress.org/page/journal/14602075/authorguide>).

- Expanded View files (replacing Supplementary Information)

We realize that it is difficult to revise to a specific deadline. In the interest of protecting the conceptual advance provided by the work, we recommend a revision within 3 months (16th Jan 2025). Please discuss the revision progress ahead of this time with the editor if you require more time to complete the revisions.

Referee #1:

The authors have thoroughly revised the manuscript and have appropriately addressed each of my initial comments. The manuscript will be crucial for the field.

Referee #2:

It looks like the authors answered my concerns

The authors addressed the remaining editorial issues.

Dear Dr. Hochberg,

I am pleased to inform you that your manuscript has been accepted for publication in the EMBO Journal.

Yours sincerely,

Yehu Moran
Academic Editor
The EMBO Journal
